# The activity-dependent histone variant H2BE modulates the life span of olfactory neurons

**Stephen W Santoro, Catherine Dulac\***

Howard Hughes Medical Institute, Department of Molecular and Cellular Biology, Harvard University, Cambridge, United States

**Abstract** We have identified a replication-independent histone variant, *Hist2h2be* (referred to herein as *H2be*), which is expressed exclusively by olfactory chemosensory neurons. Levels of H2BE are heterogeneous among olfactory neurons, but stereotyped according to the identity of the co-expressed olfactory receptor (OR). Gain- and loss-of-function experiments demonstrate that changes in *H2be* expression affect olfactory function and OR representation in the adult olfactory epithelium. We show that H2BE expression is reduced by sensory activity and that it promotes neuronal cell death, such that inactive olfactory neurons display higher levels of the variant and shorter life spans. Post-translational modifications (PTMs) of H2BE differ from those of the canonical H2B, consistent with a role for H2BE in altering transcription. We propose a physiological function for *H2be* in modulating olfactory neuron population dynamics to adapt the OR repertoire to the environment.

## Introduction

The cellular composition and connectivity of vertebrate sensory systems are shaped by signals from the external environment. These activity-dependent changes occur during critical windows of neuronal development as well as in the adult brain, enabling the animal to best perform in a given environment. Pioneering experiments in the visual system showed that patterns of light stimuli reaching each of the two eyes are essential for the activity-dependent refinement of ocular dominance columns in the visual cortex during perinatal development (*Hubel and Wiesel, 1977*). Similarly, experience-dependent plasticity has been shown to participate in the functional maturation of other sensory systems, including the auditory, somatosensory, and olfactory systems (*Hensch, 2004*). In addition to their important role in shaping sensory circuits during development, environmental stimuli can also significantly affect adult brain structures, leading to adaptive as well as maladaptive changes in sensory responses (*Buonomano and Merzenich, 1998*; *Ramachandran and Hirstein, 1998*; *Moseley and Flor, 2012*).

Experience-dependent changes in sensory systems alter the cellular composition of sensory relays, as well as the excitability and synaptic connections of neurons involved in processing sensory information. Although a molecular-level understanding of these changes is far from complete, synaptic refinement and activity-dependent transcriptional changes appear to play prominent roles (*Holtmaat and Svoboda, 2009*; *Dulac, 2010*; *Riccio, 2010*; *West and Greenberg, 2011*).

To date, activity-dependent structural remodeling of sensory systems has primarily been demonstrated in the central nervous system rather than in peripheral organs. The mouse main olfactory epithelium (MOE) offers a unique opportunity to investigate the range and mechanisms of experience-dependent plasticity within a peripheral sensory tissue, where the primary sensory detection occurs. The MOE detects large arrays of chemical cues through the expression of a large family of olfactory receptor (OR) genes (*Buck and Axel, 1991*). Individual olfactory neurons express a single OR allele

\*For correspondence: dulac@fas.harvard.edu

**Reviewing editor**: Liqun Luo, Stanford University, United States

**eLife digest** A hallmark of the nervous systems of all mammals is their capacity to undergo changes in function that are shaped by experience. This phenomenon underlies the ability of our brains to develop properly and to learn, and also enables various sensory systems—including the visual, auditory and olfactory systems—to perform optimally in diverse environments.

In most mammals, a high-functioning olfactory system is essential for carrying out tasks that are crucial for survival, such as finding food, avoiding predators and mating. In general, sensory systems have to decipher only a limited collection of stimuli, but the olfactory system must be able to process information from thousands of distinct odors that are found in a given environment and which may vary dramatically from one environment to the next. Each odor-sensing neuron in the nose of a mammal contains just one kind of odorant receptor protein, although mammalian genomes typically encode 1000 or so different kinds of receptor proteins. This suggests that it might be possible to 'tune' the olfactory system to a particular environment by changing the relative numbers of the different types of neurons. Indeed, it is known that the relative abundance of each type of odor-sensing neuron changes with age and experience, and that these changes might be caused by variations in the lifespans of the neurons.

Although our understanding of how these experience-dependent changes are orchestrated at the molecular level is far from complete, it is clear that adjustments in the levels of specific gene products is necessary. But how do experiences alter the levels of gene products to give rise to lasting changes in the brain? One hypothesis is that changes to a structure called chromatin are key to this process: chromatin is an assembly of DNA molecules, which are quite long, and organizing proteins, mostly proteins known as histones, that together form a compact structure that can fit inside the nucleus of a cell.

Santoro and Dulac have now discovered a previously uncharacterized protein called H2BE that is found only in the odor-sensing neurons of mice. H2BE is a variant of a protein called H2B, which is a well-known histone. They found that in odor-sensing neurons, H2BE replaces H2B to an extent that depends on the amount of activity experienced by the neuron: H2BE is nearly undetectable in highly active neurons, but almost completely replaces H2B in neurons that are inactive. Moreover, genetic manipulation showed that the deletion of H2BE significantly extended the lifespan of neurons, whereas elevated levels of H2BE shortened their lifespan. These findings reveal an extraordinary process that involves inactive odor-sensing neurons being depleted relative to active ones over time.

How does H2BE, which differs from H2B by just five amino acids, cause such dramatic changes in neuronal composition? One hint comes from evidence that these amino acids disrupt interactions between chromatin and 'effector' proteins, which modulate gene activity. Consistent with this, Santoro and Dulac have found that the replacement of H2B by H2BE strongly alters gene activity, although the precise mechanism by which these alterations regulate neuronal lifespans remains to be determined. Understanding this process in detail, and exploring if similar phenomena are involved in experience-dependent changes elsewhere in the nervous system, are fascinating areas of future research.

chosen through a largely stochastic process (*Chess et al., 1994*). The olfactory epithelium of mammals displays continuous neurogenesis throughout adulthood, such that a slow dividing olfactory neural stem cell population continuously replaces mature olfactory neurons that have variable, though finite, life spans (*Kondo et al., 2010*).

We identified a histone H2B variant, *H2be*, which is exclusively expressed by olfactory sensory neurons, and we hypothesized may participate in olfactory neuron gene regulation. We show that *H2be* displays activity-dependent expression, and that it regulates the transcriptional program and life span of olfactory sensory neurons. Our data suggest that *H2be* participates in a pathway that shapes the cellular and molecular composition of the olfactory epithelium based on signals from the external environment, and thus uncover a novel chromatin-based mechanism for activity-dependent neuronal plasticity.

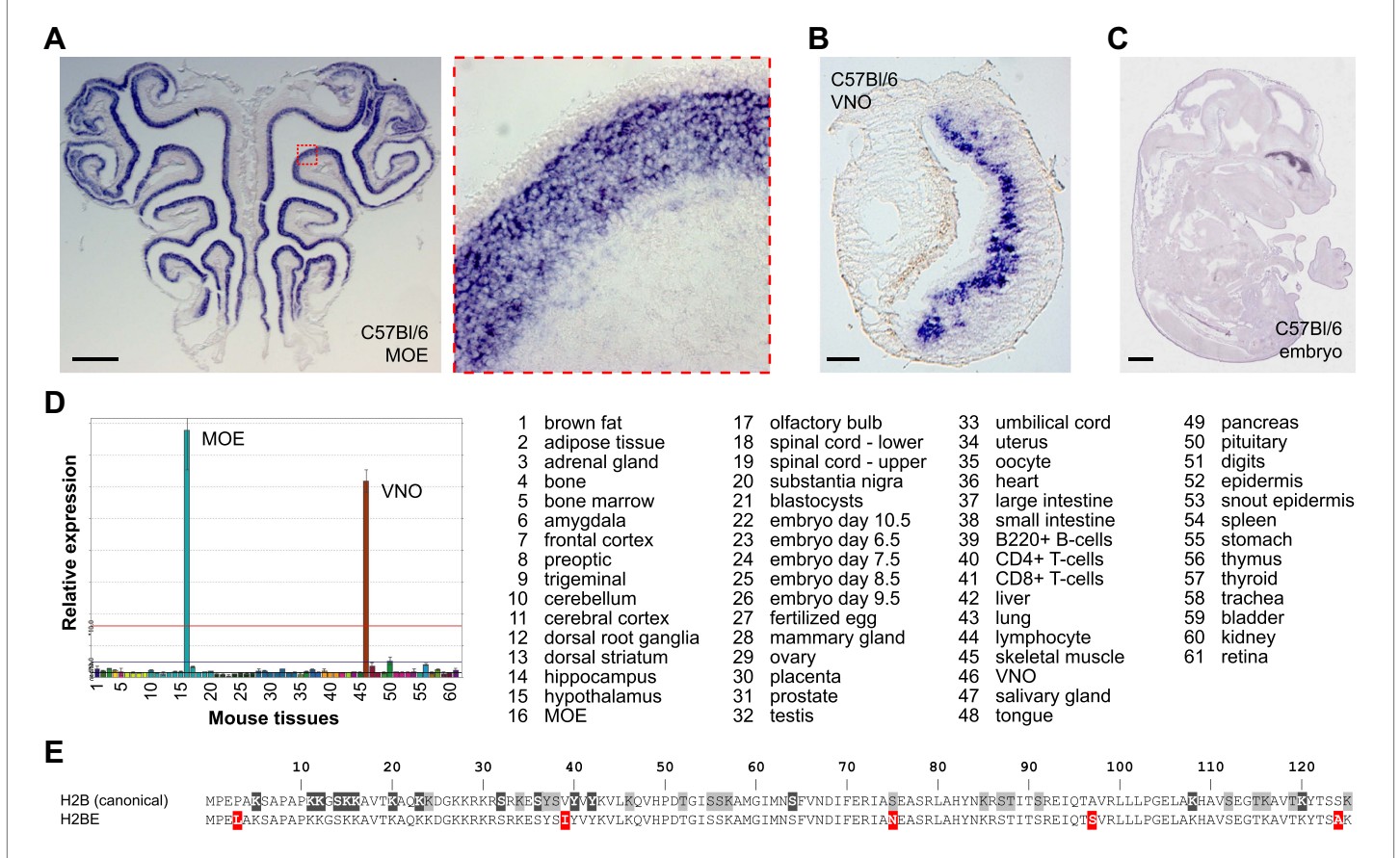

**Figure 1**. Mouse H2BE is detected exclusively in chemosensory neurons. (**A**) Analysis of *H2be* mRNA in the MOE of an 8-week old mouse, showing expression limited to sensory neurons. Boxed region is magnified (right). (**B**) Analysis of *H2be* mRNA in the VNO of an 8-week old mouse, showing expression limited to sensory neurons, especially in the apical zone. (**C**) Analysis of *H2be* mRNA in a sagittal section of an E14.5 mouse embryo (image from genepaint.org; ID: ES2590). (**D**) Profile of *H2be* mRNA levels in 61 mouse tissues (listed, right), showing exclusive expression in MOE and VNO. Data from GNF, now maintained by BioGPS (http://biogps.org/). (**E**) Alignment of H2BE and canonical H2B sequences. H2BE variant positions are highlighted in red; H2B PTM sites supported by >5 or ≤5 reports are highlighted in dark and light gray, respectively (http://www.phosphosite.org). Scale bars for (A), 500 µm; (B), 100 µm; (C), 1000 µm.

## Results

### H2BE is an olfactory-specific, replication-independent histone variant

A molecular and bioinformatics search for genes expressed differentially in neuronal subpopulations of the mouse MOE and VNO led to the identification of an uncharacterized H2B histone variant (symbol: *Hist2h2be*; named for its position (e) among H2B-encoding genes within mouse histone cluster 2 (***Marzluff et al., 2002***) and referred to herein as *H2be*). Microarray and in situ hybridization (ISH) analyses revealed high expression of *H2be* in the MOE and apical VNO neuroepithelium (***Figure 1A,B***), but no expression in the olfactory bulb (OB) or brain. The Genepaint and Genomics Institute of the Novartis Research Foundation (GNF) GeneAtlas (***Su et al., 2004***) databases, which contain transcriptional information for embryo sections (***Figure 1C***) and 61 mouse tissues (***Figure 1D***), respectively, confirmed the remarkable specificity of *H2be* expression in the MOE and VNO.

The *H2be* mRNA possesses a long 3′-untranslated region (UTR) and a poly-A tail. This contrasts with typical histone transcripts that lack poly-A tails but contain short 3′-stem loop UTRs, which facilitate coordination of histone expression with the cell cycle. These features, together with the observed presence of its mRNA in post-mitotic neurons, suggest that *H2be* encodes a replication-independent replacement histone. H2BE displays only five amino-acid differences with the canonical mouse H2B

protein (*Figure 1E*). Potential human (*Collart et al., 1992*), rat, and bovine orthologs to *H2be* exist, although their expression is uncharacterized.

To facilitate the characterization and unambiguous identification of H2BE from canonical H2B, we fused a FLAG tag to the N-terminus of H2BE, an approach used successfully for several other H2B proteins (*Kao and Osley, 2003*), and constructed a bacterial artificial chromosome (BAC) transgenic mouse line expressing the tagged protein (*Figure 2A*). Control experiments confirmed insertion of the transgene into the genome as a single copy and recapitulation of endogenous *H2be* expression (*Figure 2B–E*). We refer to the *H2be:Flag-H2be* mouse line as Flag-H2be.

## *H2be* expression levels are variable and stereotyped according to the co-expressed OR

Due to the abundance of available molecular and genetic tools, we focused our study on the prospective role of *H2be* in the MOE. Initial expression analyses revealed that *H2be* mRNA and protein levels are not uniform among olfactory neurons. Rather, neurons with variable expression levels appear intermingled, and apically-located neurons generally display the highest levels of expression (*Figures 1A and 2B*). The heterogeneous expression of *H2be* among MOE neurons could potentially reflect differences in the identity of the co-expressed OR, in neuronal maturity, or in some unknown biological feature. To examine the relationship between *H2be* and OR expression, we used fluorescent ISH (FISH) to identify mature neurons expressing each of 42 specific OR genes. We then used quantitative fluorescence microscopy (*Waters, 2009*) (see 'Materials and methods') to assess the level of FLAG-H2BE within nuclei of the identified neurons relative to their surrounding neuronal field, imaged in tissue sections spanning the length of the MOE, and across multiple animals ($n \geq 2$ per OR). Strikingly, each OR tested appears consistently and reproducibly associated with a stereotyped level of H2BE in mature neurons (*Figure 2F,G*). Indeed, for 73% of the tested ORs, neurons expressing the same OR display significantly less variation with each other in their H2BE level than is observed within the entire MOE neuronal population ($p < 0.05$ after false-discovery rate [FDR] correction; 1-tailed *F*-test). These results indicate that H2BE expression in a given neuron is tightly correlated with the identity of the co-expressed OR, and not with a specific stage of neuronal development or maturity.

## Loss of *H2be* alters olfactory function and gene expression

To further investigate the function of *H2be*, we generated an *H2be*-null line by targeted replacement of the *H2be* coding sequence with a membrane-localized mCherry reporter (*Gap43-mCherry*; *Figure 3A*). We refer to this line as H2be-KO. Analysis of GAP43-mCherry fluorescence in the MOE of H2be-KO mice revealed a variable intensity of the fluorescent reporter similar to that observed for the endogenous gene (*Figure 3B*). Analysis of GAP43-mCherry in the olfactory bulb revealed fluorescence limited to sensory axon termini in the glomerular layer. Fluorescence intensities appeared to vary widely among glomeruli, which are each innervated by neurons expressing a given OR, a result consistent with our observations of variable but stereotyped H2BE levels in neurons expressing specific ORs (*Figure 3C*).

H2be-KO mice appear healthy and fertile. Training of H2be-KO mice and littermate controls to discriminate between odors for a water reward revealed that H2be-KO mice learn the task more slowly and perform less efficiently than the controls, suggesting a defect in olfactory function (*Figure 3D*). To investigate a potential defect in olfactory signal transduction, we performed electro-olfactograms on MOEs from H2be-KO and heterozygous littermates (*Figure 3E*) but found no obvious anomalies, indicating that the observed phenotype is not due to gross defects in odor-evoked signaling.

Because histones are central components of chromatin, we suspected that loss of *H2be* might affect olfactory gene expression. Microarray analysis of whole MOE tissue in 6-month old H2be-KO and WT mice revealed that approximately 6% of transcripts are differentially expressed ($p < 0.05$; *Supplementary file 1A*; *Figure 4—source data 1*), a value that may underestimate the effects of *H2be* loss in the fraction of MOE cells that express *H2be*. Qualitatively, the effects of *H2be* loss appear to depend on the normal expression level of a given gene: highly transcribed genes tend to be upregulated, while moderately transcribed genes appear up- or down-regulated in H2be-KO mice (not shown). Although the differentially-expressed genes do not pass the $p < 0.05$ threshold after FDR adjustment for the 28,064 probe sets tested, likely due to their moderate fold changes, they do show statistically significant enrichment in several gene ontology categories (FDR-adjusted $p < 0.05$; *Figure 4A*). The categories enriched most strongly among up-regulated genes include 'olfactory

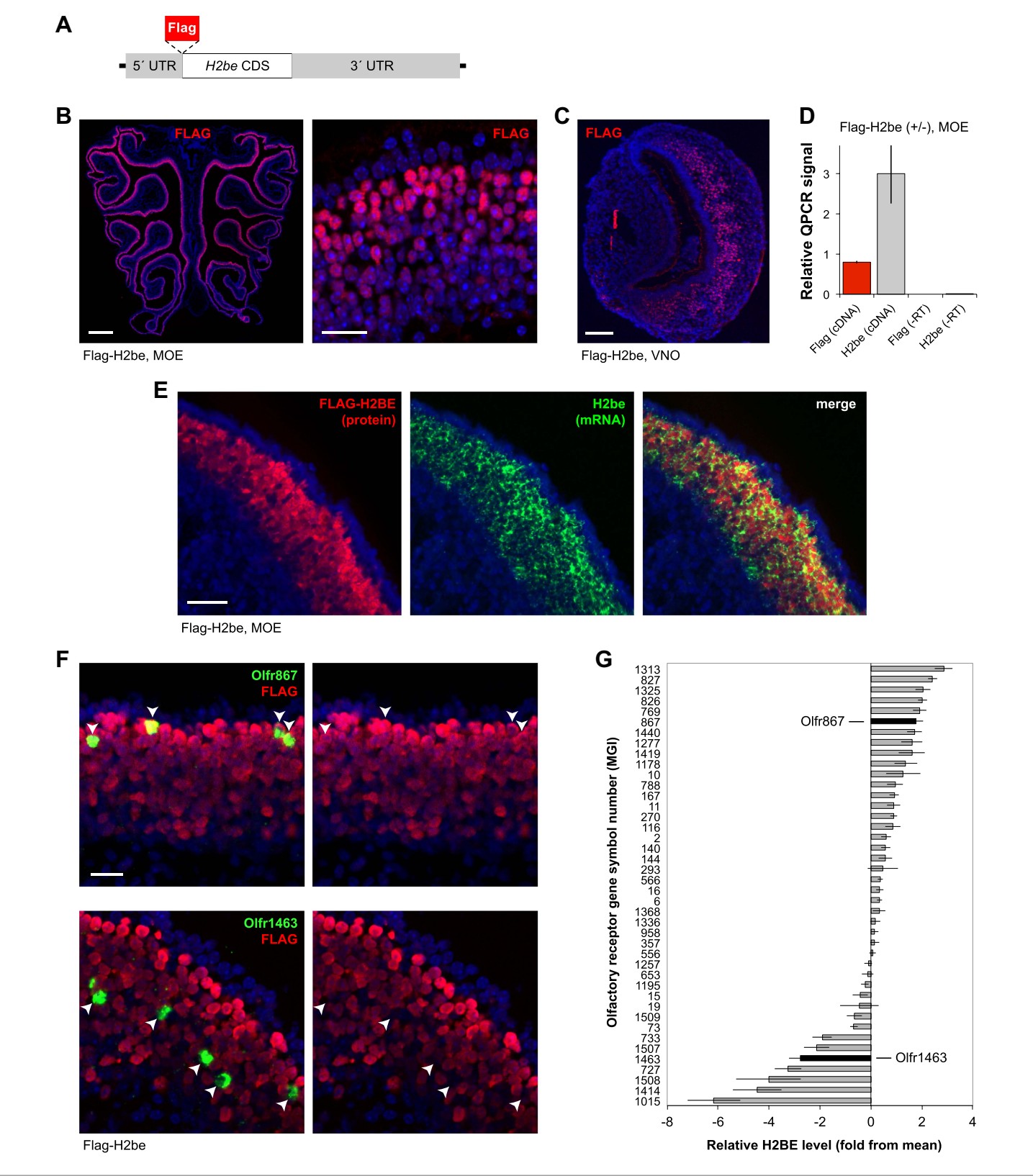

**Figure 2**. Generation of an *H2be:Flag-H2be* transgenic mouse (referred to as Flag-H2be) reveals that H2BE levels are stereotyped according to OR identity. (**A**) Flag-H2be transgenic construct, generated through modification of a BAC containing the *H2be* genomic region by insertion of a FLAG-encoding sequence immediately upstream of the *H2be* CDS. (**B,C**) Representative images of FLAG-H2BE in MOE (B) and VNO (C) from 10-week old Flag-H2be transgenic

*Figure 2. Continued on next page*

*Figure 2. Continued*

mice. (**D**) Quantitative PCR (qPCR) analysis of Flag-H2be transgene mRNA levels in whole MOE tissue from three-week old Flag-H2be$^{(+/-)}$ transgenic mice. Signals were normalized to a value of three, corresponding to a primer pair recognizing all three *H2be* mRNAs (two endogenous and one transgenic). A primer pair specific for the Flag-tagged transgenic allele produces a normalized signal of approximately 1, indicating similar per-allele expression levels for the transgenic and endogenous *H2be* alleles. Negative control samples (−RT) were prepared by omitting reverse transcriptase during cDNA synthesis. (**E**) Colocalization analysis of FLAG-H2BE protein and *H2be* mRNA in the MOE of a 3-week old Flag-H2be$^{(+/-)}$ transgenic mouse. Observation of occasional basally-located neurons that are *H2be*-mRNA-positive and FLAG-H2BE-negative is likely due to an expected lag in protein production and accumulation following *H2be* transcription onset during neuronal development. (**F**) Colocalization analysis of *Olfr867* or *Olfr1463* (arrowheads) and FLAG-H2BE, showing representative ORs associated with high or low levels of H2BE, respectively. Mouse age: 10 weeks. (**G**) Quantification of average H2BE levels in neurons expressing specific ORs (*n* = from 4 to 36 neurons per OR examined; mean, 18). Gene symbols are from the Mouse Genome Informatics database (MGI; http://www.informatics.jax.org/). Scale bars for (B, left), 500 µm; (B, right) and (F), 20 µm; (C) and (E), 100 µm.

detection' (consisting of OR genes) and 'RNA processing', and among down-regulated genes include 'developmental process' and 'positive regulation of transcription'.

The highly significant enrichment of the 'olfactory detection' gene category (FDR-adjusted p=2 × 10$^{-17}$) reveals that a large number of ORs may be up-regulated in the olfactory epithelia of H2be-KO mice, which we indeed confirmed by quantitative PCR on a subset of genes (*Figure 4B*). Interestingly, a comparison of gene expression at 6 months with an earlier age of 5 weeks revealed that OR expression differences between WT and H2be-KO mice increase dramatically with age (*Figure 4C*), and display a significant bias towards up-regulation in the KO (p<10$^{-30}$, 1-tailed paired *t*-test). To determine if defects in OR mRNA levels originate from differences in the frequency of OR expression in the MOE, in the cellular level of OR transcripts in individual neurons, or in both, we analyzed the expression of specific ORs by RNA FISH. For each of four ORs quantified, expression differences in H2be-KO and WT mice strongly correlate with differences in OR frequencies (*Figure 4D,E*) and not cellular levels (not shown). Remarkably, expression differences observed for various ORs in H2be-KO and WT mice were also found to correlate with the level of H2BE with which they are normally co-expressed: ORs with increased expression in H2be-KO mice are associated with high levels of H2BE, while ORs with decreased or unchanged expression appear associated with low H2BE levels (*Figure 4F*).

## Ectopic over-expression of *H2be* alters OR expression frequencies

Changes in the representation of neurons expressing specific ORs upon loss of *H2be* led us to predict that ectopic over-expression of *H2be* might also affect OR frequencies. We generated an *Omp:Flag-H2be* transgenic mouse line (referred to as H2be-GF) in which *Flag-H2be* is expressed under the control of the promoter for olfactory marker protein (*Omp*; *Figure 5A*), a gene highly expressed in mature olfactory neurons (*Danciger et al., 1989*). An H2be-GF founder line was identified by quantitative PCR as expressing the transgene at a approximately 10-fold higher level than the endogenous *H2be* gene (not shown). This line displayed a high level of FLAG-H2BE in all mature olfactory neurons except for a subset in zone 2 (*Figure 5B*). We reasoned that this serendipitous mosaic expression provided an internal control for the effects of *H2be* over-expression. Comparison of overall gene expression in MOE tissue from H2be-GF and WT mice revealed widespread differences (*Supplementary file 1B*; *Figure 5—source data 1*). Strikingly, both up- and down-regulated genes are dominated by ORs, exhibiting FDR-adjusted p-values of 7 × 10$^{-29}$ and 3 × 10$^{-18}$, respectively, for enrichment in the 'olfactory detection' category (*Figure 5C*). Using an FDR-adjusted p-value cutoff of 0.05, we found that approximately 7% of ORs display significant expression differences in H2be-GF mice (*Figure 5D*). As observed for H2be-KO mice, OR expression changes in H2be-GF mice are strongly correlated with H2BE expression (*Figure 5E,F,H*) and reflect changes in OR expression frequency, such that ORs that are consistently co-expressed with the transgene are reduced in frequency, while ORs that escape co-expression with the transgene within zone 2 are increased (*Figure 5F–I*).

## H2BE does not alter OR gene choice, but affects neuronal longevity

Altered OR expression frequencies in H2be-KO and H2be-GF mice could result from changes in OR gene choice or in the life span of neurons expressing a given receptor. To investigate the temporal onset of H2BE expression relative to OR gene choice, we took advantage of the basal to apical gradient of neuronal development in the MOE. Analysis of the onset of H2BE relative to ORs normally

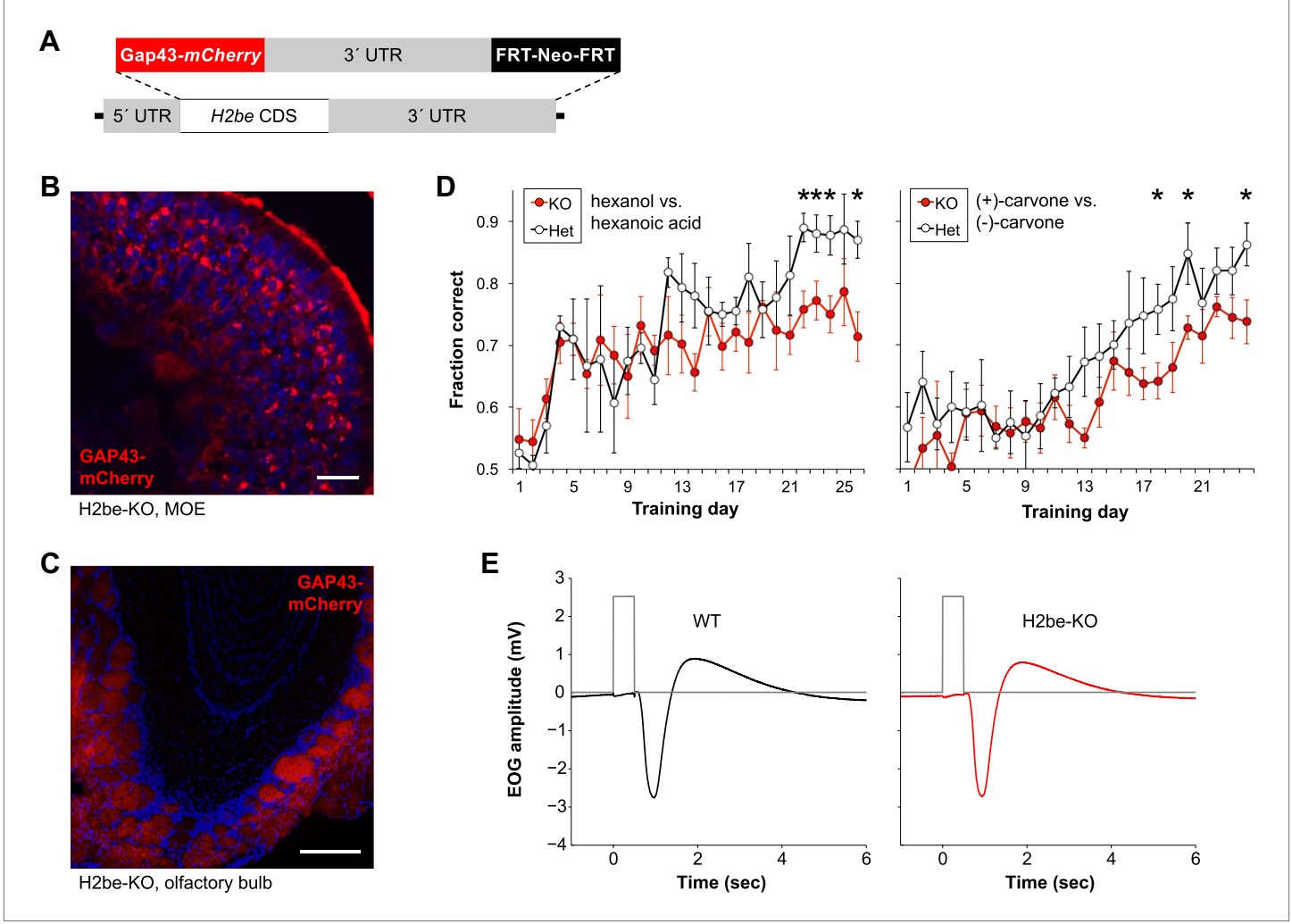

**Figure 3**. Generation of an H2be-KO/GAP43-mCherry-KI mouse line (referred to as H2be-KO) reveals that loss of *H2be* causes defects in olfaction. (**A**) H2be-KO allele, constructed through replacement of the endogenous *H2be* CDS with a membrane-targeted mCherry-encoding sequence (*Gap43-mCherry*). (**B,C**) Intrinsic GAP43-mCherry fluorescence in the MOE (B) and OB (C) of H2be-KO mice, showing GAP43-mCherry localization to the cell membranes and processes of olfactory neurons. Mouse ages: (B), 6 months; (C), 2 months. Scale bar for (B), 20 μm; (C), 200 μm. (**D**) Performance of approximately 3-month old water-restricted H2be-KO and control littermates in discriminating between hexanol/hexanoic acid (left) or (+)/(−)-carvone (right) odor pairs to obtain water (*n* = 5 per genotype). *p<0.05. (**E**) Effects of *H2be* loss-of-function on odor-evoked electrical responses in the MOE. Electro-olfactogram traces (black and red) represent average responses to a 0.5-s stream of air from the head space of a 1% solution of isoamyl-acetate in mineral oil. Gray traces show timing of switching between the delivery of clean, de-odorized air (low), and odor-containing air (high). Results shown are representative of multiple trials, odorants, and concentrations; experimental procedures were adapted from those described previously (*Waggener and Coppola, 2007*).

associated with high levels of H2BE revealed that H2BE expression is initiated well after OR expression (*Figure 6A*). We next compared the onset of *H2be* expression to that of *Neurod1*, *Gap43*, and *Omp*, which are expressed prior to, concurrently with, and subsequent to OR gene choice, respectively (*Cau et al., 2002*; *Iwema and Schwob, 2003*; *Kolterud et al., 2004*). We found that the onset of *H2be* expression follows the window of *Neurod1* expression, as observed at both the protein and RNA levels in MOE tissue from Flag-H2be and WT mice, respectively (*Figure 6B,C*). Moreover, H2BE expression begins after GAP43 (*Figure 6D*) and before OMP (*Figure 6E*), but overlaps extensively with both. Together, these results confirm that H2BE is present prior to full neuronal maturity, but subsequent to OR choice. Thus H2BE is unlikely to participate in the process of OR choice.

Next, we further analyzed H2be-KO mice for evidence of defects in choice, or in stabilization of OR gene choice. We found no defects in OR co-expression frequencies (*Figure 7A*) or in the position or number of glomeruli (*Figure 7B*) in H2be-KO mice, which are phenotypes expected to be associated

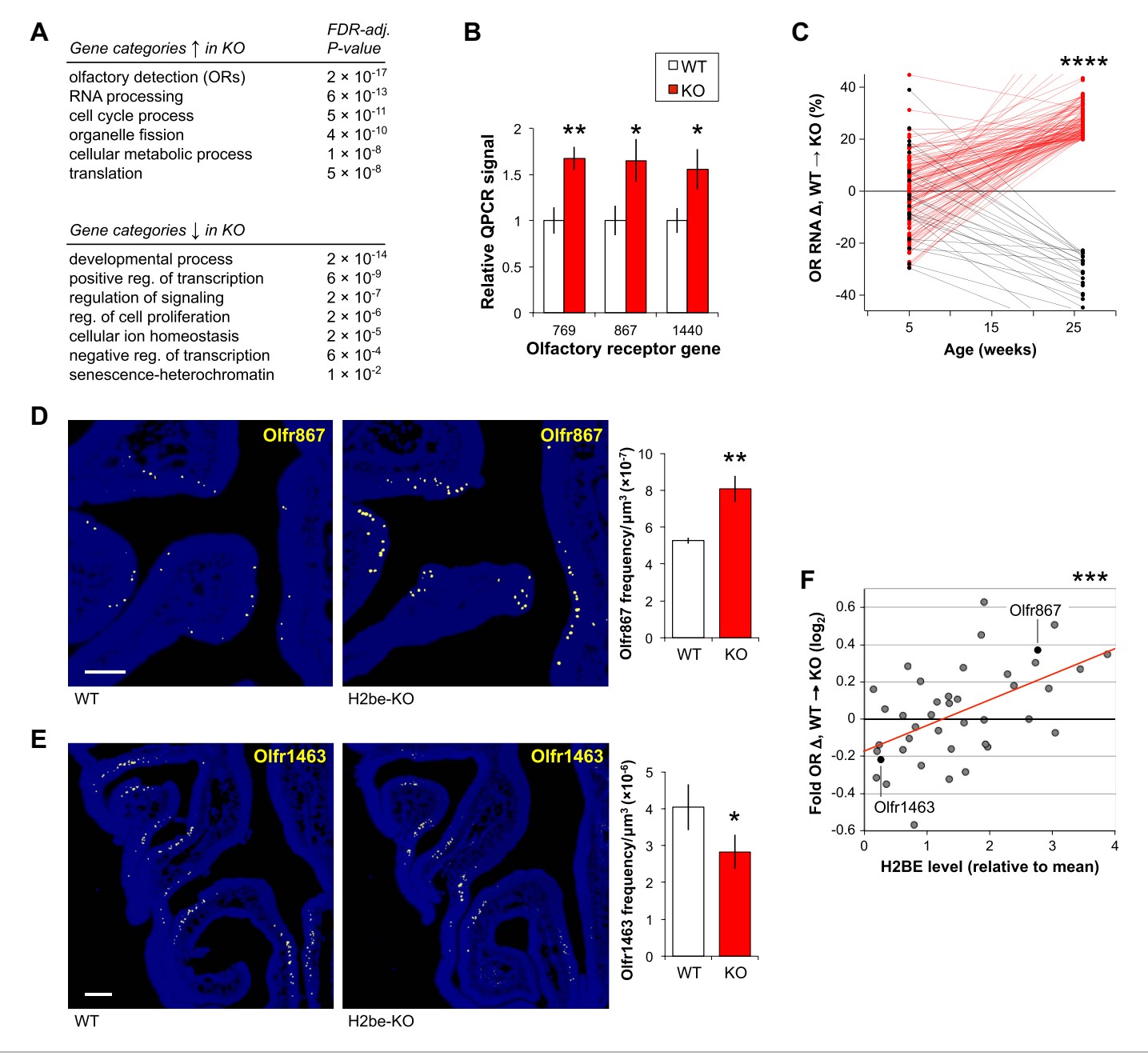

**Figure 4**. Loss of *H2be* causes defects in gene expression and OR expression frequencies. (**A**) Enriched gene ontology categories among genes up- (top) or down-regulated (bottom) in 6-month old H2be-KO vs WT MOEs, based on microarray analyses of whole MOE tissue (*n* = 6 samples/ genotype, 2 animals/sample). (**B**) Multiplex qPCR analysis of OR mRNA levels in MOE tissue of six-month old H2be-KO and WT mice. Signals represent normalized ratios of specific OR mRNAs to *Cnga2*, which is unaltered in H2be-KO mice and used as an internal control (*n* = 6). (**C**) Expression differences in H2be-KO and WT MOEs (based on microarray analysis), plotted as a function of age. For simplicity, only ORs with differences >20% at age 6 months are shown, with up- and down-regulated ORs shown in red and black, respectively; statistical analysis included all ORs interrogated. (**D**,**E**) Representative images (left) and quantification (right) of *Olfr867* (D) and *Olfr1463* (E) expression frequencies in 6-month old H2be-KO and WT littermates (*n* = 3 mice, 10 sections per mouse). Scale bars, 200 μm. (**F**) Relationship between OR gene expression defects in 6-month old male H2be-KO mice (based on microarray analysis) and stereotypical H2BE levels as measured in male Flag-H2be mice. Red line, best fit. *p<0.05, **p<0.01, ***p<0.001, ****p<10⁻³⁰.

The following source data are available for figure 4.
**Source data 1.** Effects of *H2be* loss of function on gene expression in the main olfactory epithelium of 6-month old mice.

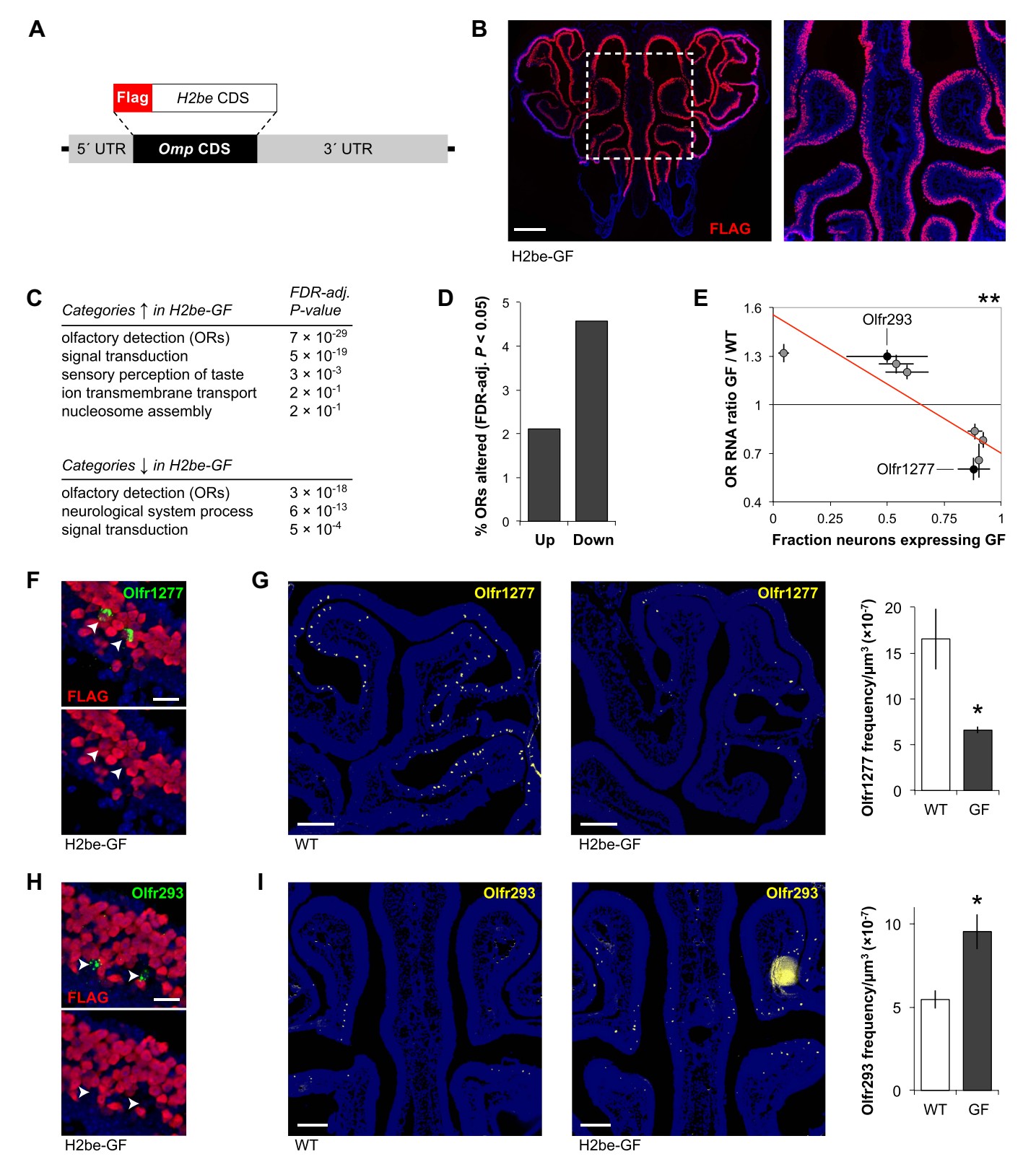

**Figure 5**. Generation of an *Omp:Flag-H2be* transgenic mouse (referred to as H2be-GF) reveals that ectopic overexpression of *H2be* in olfactory neurons alters gene expression and OR expression frequencies. (**A**) H2be-GF transgenic construct, generated through replacement of the *Omp* CDS with a FLAG-H2BE-encoding sequence in a vector containing the *Omp* genomic region. (**B**) Analysis of FLAG-H2BE in the MOE of a 5-week old H2be-GF
*Figure 5. Continued on next page*

*Figure 5. Continued*

mouse, showing high transgene expression in all mature neurons, except for a band near zone 2 (*Sullivan et al., 1996*). Boxed region is magnified (right). (**C**) Gene ontology categories enriched among genes up- (top) or down-regulated (bottom) in 5-week old H2be-GF vs WT MOEs, based on microarray analyses of whole MOE tissue (*n* = 4 samples, 3 animals per sample) and WT (*n* = 6 samples, 2 animals per sample) mice. (**D**) Percentage of OR genes with significantly differential expression (FDR-adjusted p<0.05) in 5-week old H2be-GF and WT mice. (**E**) Relationship between OR gene expression defects in H2be-GF mice and transgene co-expression penetrance. Red line, best fit. (**F,H**) Colocalization of *Olfr1277* or *Olfr293* (arrowheads) and FLAG-H2BE in 5-week old H2be-GF mice, showing representative ORs associated with high- or low transgene penetrance, respectively. (**G,I**) Representative images (left) and quantification (right) of *Olfr1277* (G) and *Olfr293* (I) expression frequencies in 5-week old H2be-GF and WT littermates (*n* = 3 mice; 10 sections per mouse). *p<0.05, **p<0.01. Scale bars for (B), 500 μm; (F) and (H), 20 μm; (G) and (I), 200 μm.
The following source data are available for figure 5.
**Source data 1.** Effects of the ectopic over-expression of *H2be* (expressed from an Omp-promoter-driven transgene and tagged with a FLAG epitope) on gene expression in the main olfactory epithelium of 5-week old mice.

with the aberrant expression of multiple OR genes or the abnormal switching between distinct ORs (*Shykind et al., 2004*), suggesting that H2BE is not likely involved in the maintenance of OR gene choice. Finally, as described previously, gene expression analyses indicate that OR expression defects in H2be-KO mice are markedly more severe at 6 months of age than at 5 weeks (*Figure 4C*). Since disruption in OR gene choice would be expected to cause defects in OR expression that are independent of age, the delayed appearance of OR expression defects in H2be-KO mice is inconsistent with a role for H2BE in this process.

We next investigated the alternative hypothesis that altered OR frequencies in H2be-KO and H2be-GF mice may result from changes in neuronal longevity. Analysis of active-CASP3-labeled mature olfactory neurons in H2be-KO and littermate controls revealed a 45% lower frequency of apoptosis in the mutant epithelium (*Figure 7C*). Moreover, analysis of the life span of 5-bromo-2′-deoxyuridine (BrdU)-labeled neurons revealed a significantly higher survival rate in H2be-KO mice compared to controls (*Figure 7D*). These results indicate that the loss of *H2be* increases the life span of mature neurons. Accordingly, similar analyses in H2be-GF mice revealed a twofold higher rate of apoptosis among mature neurons (*Figure 7E*) and a significantly reduced survival rate compared to Flag-H2be mice (*Figure 7F*). Remarkably, the initial BrdU labeling frequency of H2be-GF neurons is approximately 2.9-fold higher than that observed in control neurons, reflecting an elevated rate of neurogenesis, presumably a consequence of the increased rate of apoptosis (*Figure 7G*). To ensure that the observed increase in apoptosis in transgenic epithelia is not due to a non-specific toxic effect of the overexpressed histone, we expressed *H2be* or *Flag-H2be* at high levels in cultured fibroblast (NIH-3T3) or embryonic kidney (HEK-293T) cells. Transgene-expressing cells showed no increase in cell death compared to control cells overexpressing the canonical *H2b* or containing no transgene (not shown), suggesting that H2BE's action in affecting cellular longevity requires an olfactory-specific cellular context. Taken together, these data indicate that the expression of *H2be* in mature olfactory neurons directly affects neuronal longevity, such that neurons with low expression of H2BE tend to have longer life spans, while neurons with elevated H2BE tend to be relatively short-lived.

## H2BE levels are regulated by neuronal activity

We next sought to identify the source of heterogeneity in H2BE level among neurons expressing different ORs in WT mice. To investigate a possible link with neuronal activity, we performed unilateral naris occlusion (UNO) by surgically blocking airflow into one nostril of Flag-H2be mice for 10 days. Upon analysis of H2BE in the MOE of these mice, we observed dramatically higher H2BE expression on the closed side of the epithelium relative to the open side. This increase can be observed at the cellular level in neurons expressing a specific OR (*Figure 8A,B*), and in the GAP43-mCherry fluorescence within the ipsilateral and contralateral olfactory bulbs of H2be-KO$^{(+/-)}$ mice (*Figure 8C*). Together, these results indicate that olfactory deprivation causes an up-regulation of H2BE.

To test whether neuronal stimulation results in reduced H2BE levels, we exposed mice to a mixture of odorants or, as a negative control, to mineral oil. The odorant mixture consisted of four ligands corresponding to known ORs: heptanal/octanal for *Olfr2* (*Bozza et al., 2002*), lyral for *Olfr16* (*Touhara et al., 1999*), and eugenol for *Olfr73* (*Oka et al., 2004*) and *Olfr958* (*Oka et al., 2006*). After 3 weeks

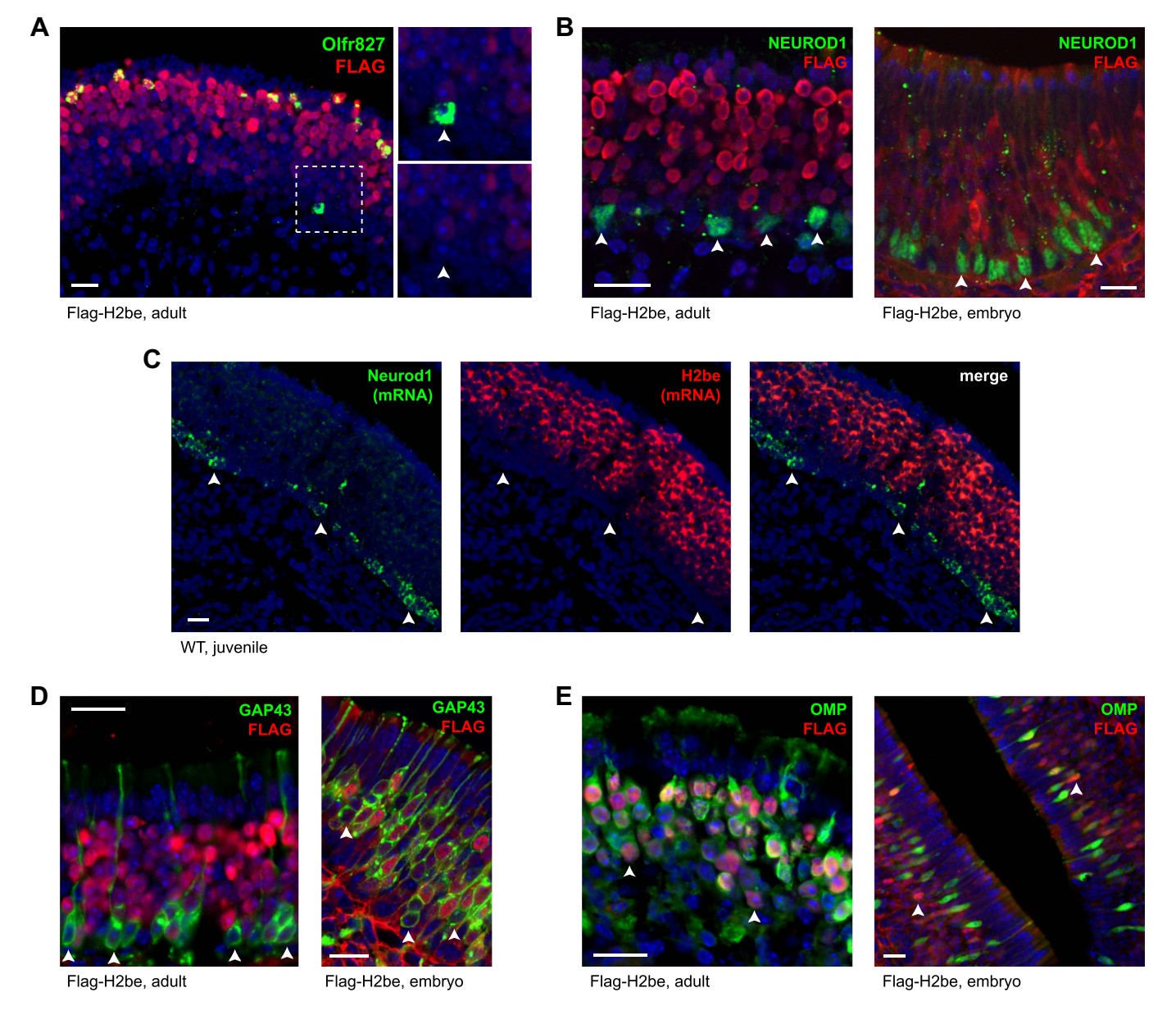

**Figure 6**. The expression onset of *H2be* follows OR choice during neuronal development. (**A**) H2BE is undetectable in an *Olfr827*⁺ neuron that is newly-differentiated (note its basal position in the epithelium; boxed and magnified, right), but expressed at uniformly high levels in mature *Olfr827*⁺ neurons. (**B,D,E**) MOE expression patterns in adult (left) or embryo (right) of *H2be* relative to *Neurod1*, *Gap43*, and *Omp*, which have expression onsets prior to, concurrently with, and following OR choice, respectively. All NEUROD1⁺ cells (B) and a fraction of basal GAP43⁺ (newly-differentiated) neurons (C) are H2BE⁻ (arrowheads), whereas a fraction of H2BE⁺ neurons are OMP⁻ (E; arrowheads). (**C**) Colocalization of endogenous *H2be* and *Neurod1* mRNAs in the MOE of a 3-week old WT mouse. Mouse ages: (A), 3 months; (B, left), 4 months; (C), 3 weeks; (D, left) and (E, left), 10 months. Scale bars for (A) to (E), 20 µm.

of exposure to the odor mixture or mineral oil, the four stimulated neuronal subtypes showed significantly reduced levels of H2BE compared to the same subtypes in control mice (*Figure 8D–F*; *Figure 8—figure supplement 1*). In contrast, neurons expressing two randomly picked orphan ORs, *Olfr167* and *Olfr653*, showed no statistically significant differences between odor-exposed and control mice. Thus, olfactory stimulation of specific neurons results in a relative reduction in their nuclear H2BE levels.

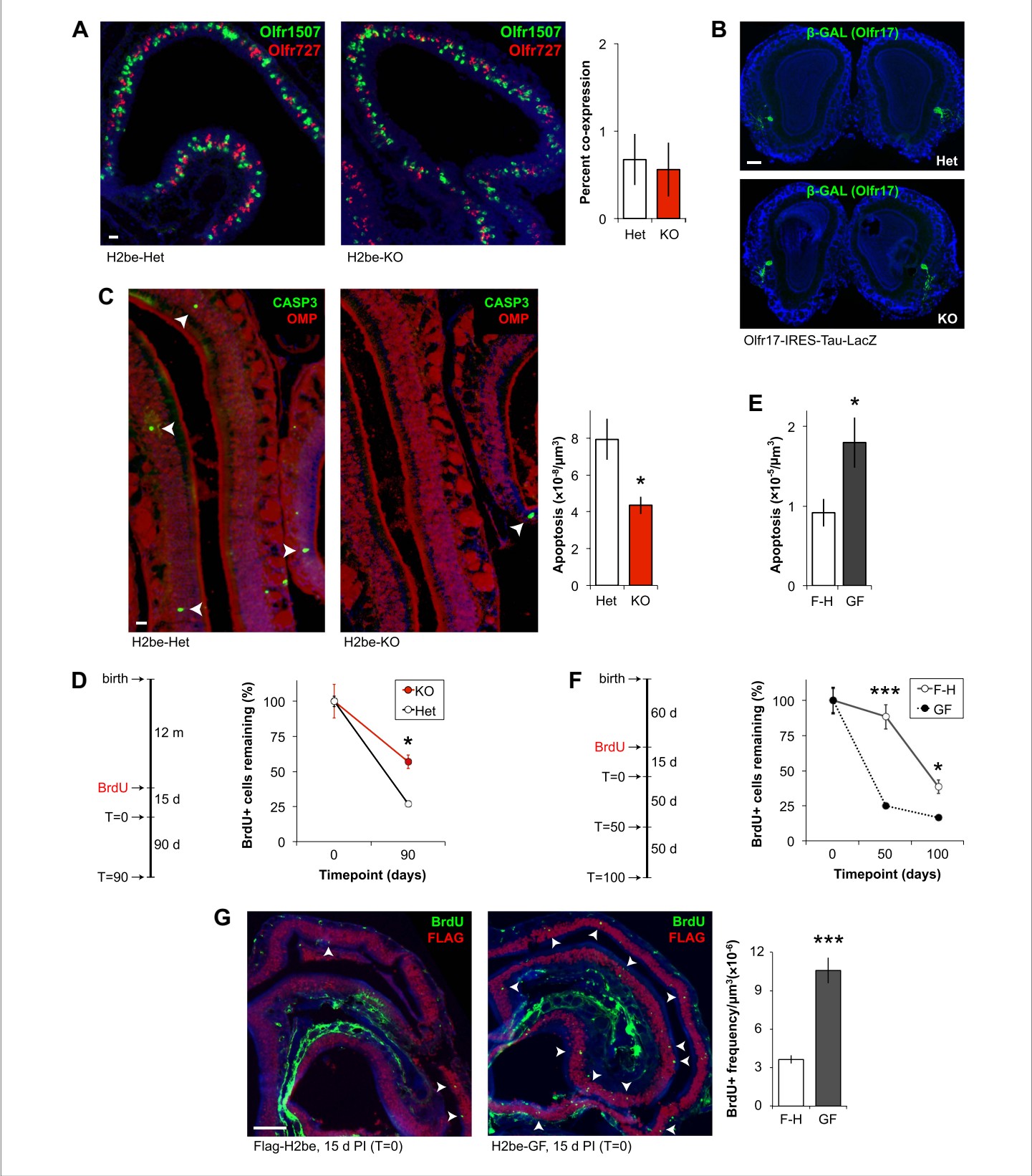

**Figure 7**. H2BE affects olfactory neuronal longevity, not OR choice. (**A**) Representative images (left) and quantification (right) of *Olfr1507* and *Olfr727* co-expression frequencies in H2be-KO and control littermates (*n* = 10 sections). (**B**) Labeled axons of *Olfr17*-expressing neurons form indistinguishable
*Figure 7. Continued on next page*

*Figure 7. Continued*

glomeruli in the OBs of H2be-KO and control littermates. (**C,E**) Representative images (C, left; arrowheads) and quantification (C, right; E) of apoptosis in mature neurons of H2be-KO (C) or H2be-GF (E) mice compared to controls (C: *n* = 3 mice, 12 sections per mouse; E: *n* = 10 sections). (**D,F**) Schematic of experimental analysis timeline (left) and quantification (right) of relative BrdU+ neuron frequencies in H2be-KO (D) and H2be-GF (F) mice compared to controls, respectively (D: *n* = 3 mice per timepoint, 12 sections per mouse; F: *n* = 10 sections per timepoint). F-H: Flag-H2be. (**G**) Representative images (left) and quantification (right) of neurogenesis in H2be-GF (GF) mice and Flag-H2be (F-H) controls by analysis of frequencies of BrdU+ neurons (arrowheads) 15 days post-injection of BrdU (*T* = 0 timepoint; *n* = 10 sections). *p<0.05, ***p<0.001. Mouse ages: (A) and (E), 4 months; (B) and (G), 2 months; (C), 15 months. Scale bars for (A) and (C), 20 µm; (B), 200 µm; (G), 100 µm.

To further corroborate the activity-dependence of H2BE, we used GAP43-mCherry fluorescence as a reporter of *H2be* gene expression in H2be-KO$^{(+/−)}$ mice and measured its intensity relative to that of tyrosine hydroxylase, a marker of neuronal activity in the glomerular layer of the olfactory bulb (*Cho et al., 1996*). These analyses revealed a significant negative correlation between the two markers, further supporting the view that *H2be* gene expression is inversely related to neuronal activity (*Figure 8G,H*).

To investigate the possibility that H2BE expression is regulated by cAMP or Ca$^{2+}$, two important signaling components and modulators of gene expression in the MOE (*Mori and Sakano, 2011*), we looked for evidence of changes in H2BE levels in *Adcy3*-null and *Cnga2*-null mice, respectively. We found that olfactory neurons that contain undetectable levels of H2BE in WT mice frequently show extremely high levels in their *Adcy3*$^{(−/−)}$ counterparts (*Figure 9A*). However, using GAP43-mCherry as a reporter of *H2be* expression in *Cnga2*$^{(+/−)}$ mosaic females, we observed no elevation in GAP43-mCherry fluorescence in *Cnga2*$^{−}$ glomeruli relative to *Cnga2*$^{+}$ glomeruli (*Figure 9B*). These results suggest that the activity-dependent down-regulation of H2BE is cAMP- but not Ca$^{2+}$-mediated. The precise mechanism of H2BE regulation by cAMP is unclear. One possibility is that *H2be* expression is suppressed by a factor such as ICER, a cAMP-dependent repressor form of the cyclic-AMP responsive element binding protein (CREB) family member CREM (*Lyons and West, 2011*). Interestingly, we identified two CRE half-sites within the *H2be* coding region, although their function, if any, is unknown.

### *H2be* mediates activity-dependent changes in olfactory neurons

To investigate a potential role for *H2be* in mediating activity-dependent changes in gene expression, we performed UNO on H2be-KO and WT mice and compared gene expression differences for each genotype on the open and closed sides of the MOE after 3 weeks of nostril closure. Consistent with a recent study (*Coppola and Waggener, 2011*), we identified widespread differences in gene expression after activity deprivation in WT mice, with 'olfactory detection' (ORs) identified as the most highly enriched gene ontology category among both up- and down-regulated genes (*Figure 10A*; *Figure 10—source data 1*). Using an FDR-adjusted p-value cutoff of 0.05, approximately 11.5% of ORs display significant expression differences after olfactory deprivation in WT mice (*Figure 10B*). FISH analyses revealed that these differences reflect altered OR expression frequencies within the MOE (*Figure 10C,D*).

We then investigated a possible relationship between H2BE levels and changes in OR expression after UNO in WT mice. Remarkably, this analysis revealed that ORs that are down-regulated following UNO are normally co-expressed with high levels of H2BE, while up-regulated ORs are normally co-expressed with low H2BE levels (*Figure 10E,F*). Thus, if H2BE level is taken as a measure of neuronal inactivity, these data suggest that olfactory deprivation causes normally inactive neurons to decrease in relative abundance, while leading to a relative increase in neurons that are highly active.

Comparison of MOE expression data obtained for WT and H2be-KO mice revealed that for a subset of UNO-altered genes (*Figure 11A*; *Figure 10—source data 1*), including a large fraction of ORs (*Figure 11B,C*), UNO-mediated changes are significantly attenuated in the MOE of the KO compared to the WT. These results suggest that H2BE participates in activity-dependent modulation of olfactory gene expression, though it is clearly not the only mediator of the observed changes.

Our findings support a model in which the level of olfactory activity of a neuron determines its level of H2BE, which in turn affects its life span (*Figure 12*). According to this model, inactive neurons express high levels of H2BE, which leads to reduced longevity, while active neurons maintain

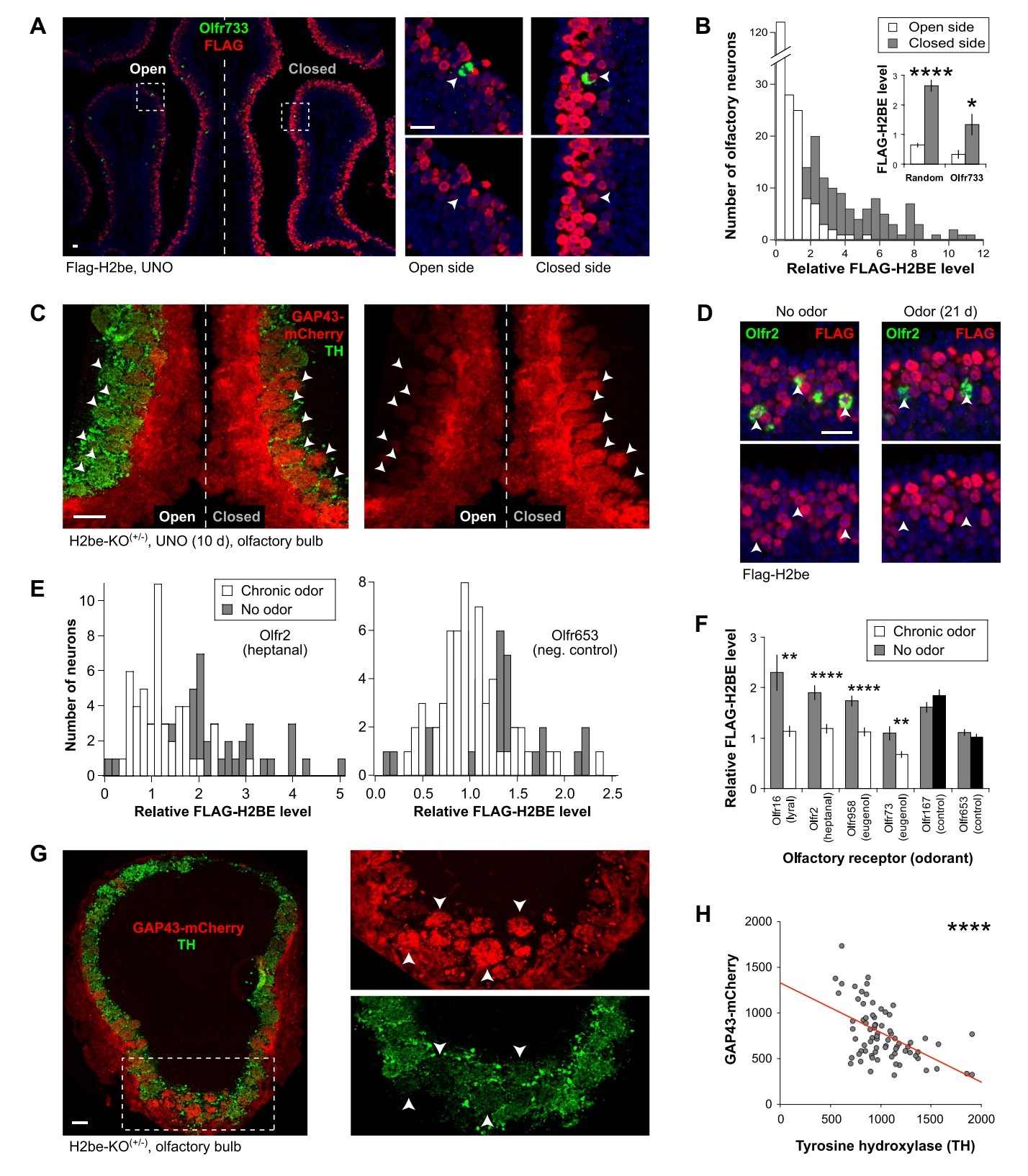

**Figure 8**. *H2be* is regulated by activity. (**A**,**B**) Effects of unilateral naris occlusion (UNO) on H2BE level in the MOE. (**A**) Representative images of FLAG-H2BE and *Olfr733* colocalization (boxed regions magnified, right; *Olfr733*+ neurons, arrowheads). (**B**) Distributions (main) and averages (inset) of H2BE level within nuclei of randomly-sampled (main; inset, left; *n* = 200) or *Olfr733*+ (inset, right; *n* = 10) neurons on the two sides of the MOE. (**C**) Effects

*Figure 8. Continued on next page*

*Figure 8. Continued*

of UNO on intrinsic GAP43-mCherry fluorescence in glomeruli (arrowheads) within the OB of an H2be-KO heterozygous mouse. Reduced tyrosine hydroxylase (TH; a marker of olfactory activity) staining on the closed side indicates completeness of naris closure. (**D–F**) Effects of odor exposure on H2BE level in olfactory neurons. Representative images (D) and quantification (E, left) of FLAG-H2BE in Olfr2$^+$ neurons (D, arrowheads) exposed to odors or mineral oil (no odor). (E, right) Quantification of FLAG-H2BE in Olfr653$^+$ neurons (chosen randomly as a negative control) exposed to odors or mineral oil (no odor). (**F**) Average H2BE level within neurons expressing odor-stimulated or control ORs ($n$ = 20–60). (**G,H**) Representative image (G; boxed region magnified, right) and quantification (H) of the relationship between GAP43-mCherry and tyrosine hydroxylase intensities within glomeruli of an H2be-KO heterozygous mouse. Red line, best fit. *p<0.05; **p<0.01; ****p<0.0001. Mouse ages: (A) and (C), 4 weeks; (D), 8 weeks; (G), 10 weeks. Scale bars for (A) and (D), 20 μm; (C) and (G), 100 μm.

The following figure supplements are available for figure 8.

**Figure supplement 1**. Distribution of relative H2BE levels within neurons expressing Olfr73, Olfr958, Olfr16, and Olfr167 for odor-exposed and control littermates. Stimulating odors are indicated, except for Olfr167, which serves as a negative control.

low levels of H2BE and are relatively long-lived. This model predicts that OR expression frequencies should change with age and experience and lead to the gradual enrichment of active olfactory neurons at the expense of inactive neurons in the MOE. Consistent with this prediction and previous reports of age-dependent changes in OR expression (*Lee et al., 2009*; *Rimbault et al., 2009*; *Rodriguez-Gil et al., 2010*), we found that 65% of all ORs are significantly altered in their expression from 5 weeks to 6 months of age in WT mice (FDR-adjusted p<0.05; not shown). Future experiments will determine the physiological significance of the ORs displaying age-dependent altered expression.

## H2BE is widely distributed on genomic DNA

To further investigate H2BE's function at the molecular level, we first sought confirmation that transgenic FLAG-H2BE protein, a tool critical for these studies, is normally integrated into chromatin. Fractionation of unfixed MOE cell nuclei from Flag-H2be and H2be-GF transgenic mice showed that FLAG-H2BE is present at barely detectable levels in soluble nucleoplasm and is instead almost entirely chromatin-bound (*Figure 13A*). Moreover, mass spectrometric analysis of proteins associated with FLAG-H2BE mononucleosomes from Flag-H2be MOE tissue identified numerous chromatin-associated proteins, including several canonical and variant histones (*Figure 13B*). Finally, chromatin immunoprecipitation (ChIP) using anti-FLAG antibodies yielded large quantities of DNA from Flag-H2be MOE tissue (*Figure 13C*). Together, these results provide strong evidence that FLAG-H2BE is readily integrated into the chromatin of olfactory neurons and support the use of Flag-H2be and H2be-GF mice for investigating the molecular function of H2BE.

Using FLAG-H2BE as a molecular surrogate for H2BE, we examined the histone variant's localization on genomic DNA. High-resolution confocal imaging of neuronal nuclei in the MOE of Flag-H2be mice revealed that H2BE is not confined to nuclear puncta, but rather is widely distributed throughout the nucleus (*Figure 13D*). Analysis of H2BE localization by genome-wide chromatin immunoprecipitation (ChIP) in MOE neurons confirmed its wide distribution and revealed a slight enrichment near gene promoters, especially in genes highly expressed in mature olfactory neurons (*Figure 13E*). Interestingly, these analyses and subsequent qPCR experiments revealed that the protein-coding regions of histone genes are particularly enriched for H2BE, while those of OR and vomeronasal receptor (VR) genes are relatively devoid of the variant (*Figure 13F–H*). Because the ChIP input was obtained from a heterogeneous mixture of MOE nuclei and approximately 99.9% of OR alleles and all VR alleles are silent in a given olfactory neuron, the latter results may reflect the lower accessibility of silent OR and VR loci for replacement of canonical H2B by H2BE, perhaps due to the highly compacted nature of these loci (*Magklara et al., 2011*). Together, our results support a model in which H2BE replacement of canonical H2B within olfactory chromatin is widespread, but most extensive within transcriptionally active loci.

## H2BE and canonical H2B display different post-translational modifications

Four of the five amino acid variant positions within H2BE are located near post-translational modification (PTM) sites of canonical H2B (*Figure 1E*), raising the question of whether PTMs may differ between

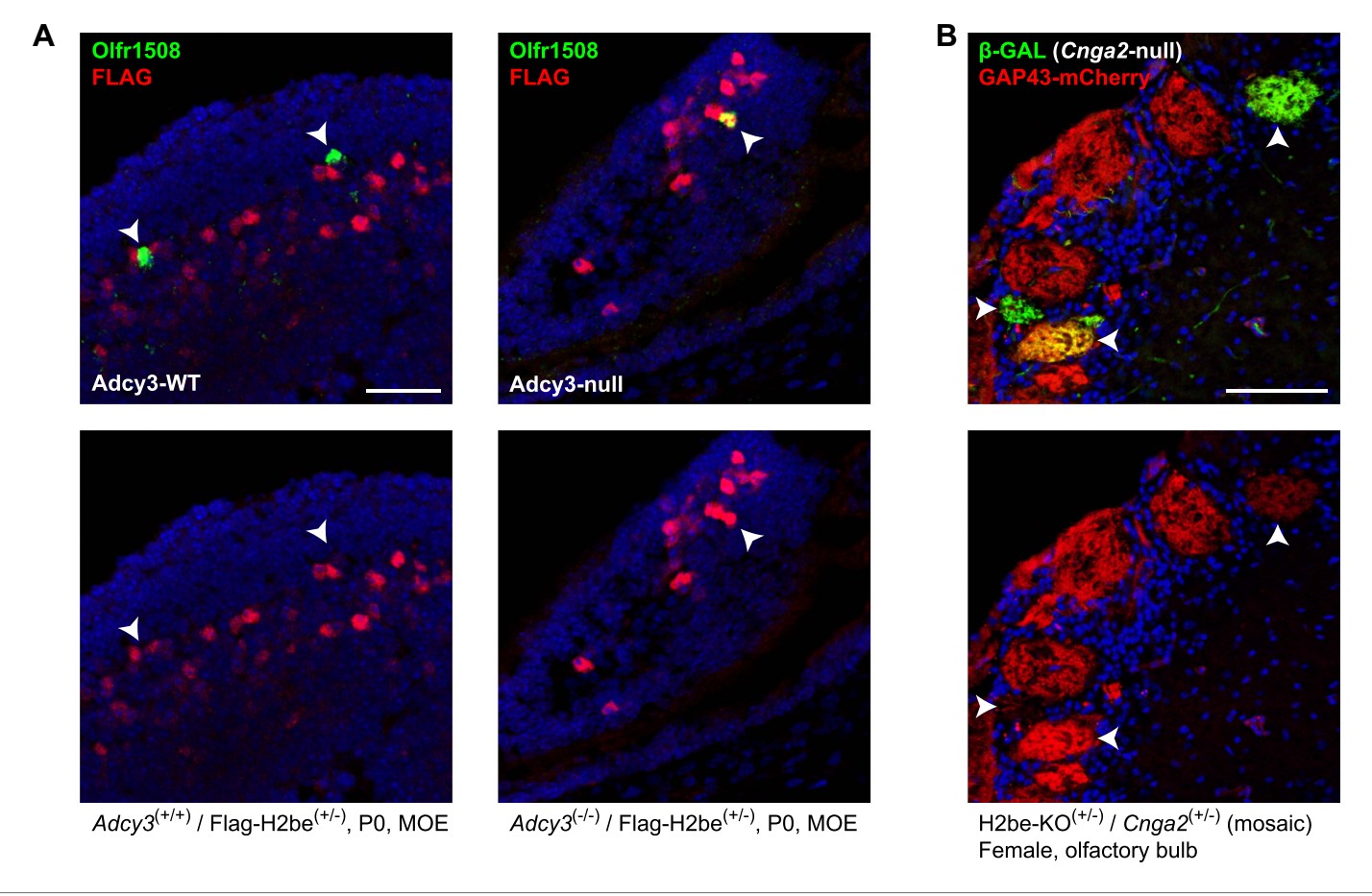

**Figure 9**. H2BE levels are cAMP-, but not $Ca^{2+}$-dependent. (**A**) Representative example of the effects of *Adcy3* loss-of-function on H2BE levels in newborn (P0) mice. In *Adcy3*[(+/+)] mice, *Olfr1508*[+] neurons contain extremely low H2BE levels (left, arrowheads), while in *Adcy3*[(−/−)] mice, *Olfr1508*[+] neurons frequently contain extremely high levels (right, arrowhead), indicating that cAMP participates in the negative regulation of H2BE. Note: age P0 was chosen due to the low postnatal survival rate of *Adcy3*[(−/−)] mice. (**B**) Effects of *Cnga2* loss-of-function on *H2be* expression. Mice in which *Cnga2* (an X-chromosomal gene necessary for odor-evoked $Ca^{2+}$ signaling) was replaced with the *Tau-LacZ* gene (***Zhao and Reed, 2001***) were crossed with H2be-KO mice to generate three-week old Cnga2-KO-Tau-LacZ[(+/−)]/H2be-KO-Gap43-mCherry[(+/−)] compound heterozygous females. In these mice, one half of new olfactory neurons express TAU-LACZ instead of CNGA2 and project to glomeruli distinct from neurons expressing *Cnga2* (***Zheng et al., 2000***). Analysis of β-GAL[+] (TAU-LACZ) and GAP43-mCherry intensities within glomeruli revealed that *Cnga2*[−] neurons (β-GAL[+]; arrowheads) do not have higher levels of GAP43-mCherry than Cnga2[+] neurons (β-GAL[−]), indicating that *H2be* expression is not negatively regulated by $Ca^{2+}$ signaling. Scale bars for (A), 40 μm; (B), 100 μm.

the two proteins. We analyzed three relatively well-characterized H2B PTMs: mono-methylation and acetylation of lysine 5 (Lys5-Me and Lys5-Ac), which lies close to the H2BE variant residue P3L, and ubiquitination of lysine 120 (Lys120-Ub), near the variant residue S124A. Both H2B-Lys5 PTMs have been shown to be positively correlated with transcriptional activity, with Lys5-Me and Lys5-Ac enriched within transcribed and promoter regions, respectively (***Barski et al., 2007***; ***Wang et al., 2008***). H2B-Lys120 ubiquitination, which has also been linked with transcription, appears to play a complex regulatory role (***Chandrasekharan et al., 2010***).

Analysis of H2BE and H2B-Lys5-Me levels in Flag-H2be MOEs revealed a striking inverse correlation between the two staining patterns, such that neurons expressing a high level of H2BE display conspicuously low H2B-Lys5-Me immunoreactivity, and vice versa (***Figure 14A,B***). Using an ELISA assay, we ruled out the possibility that the polyclonal antibody against H2B-Lys5-Me is incompatible with the H2BE sequence (***Figure 14C***). To determine if there is a causal link between the seemingly mutually exclusive expression of H2BE and H2B-Lys5-Me, we examined the prevalence of H2B-Lys5-Me in H2be-KO and H2be-GF mice. Remarkably, loss of *H2be* leads to widespread

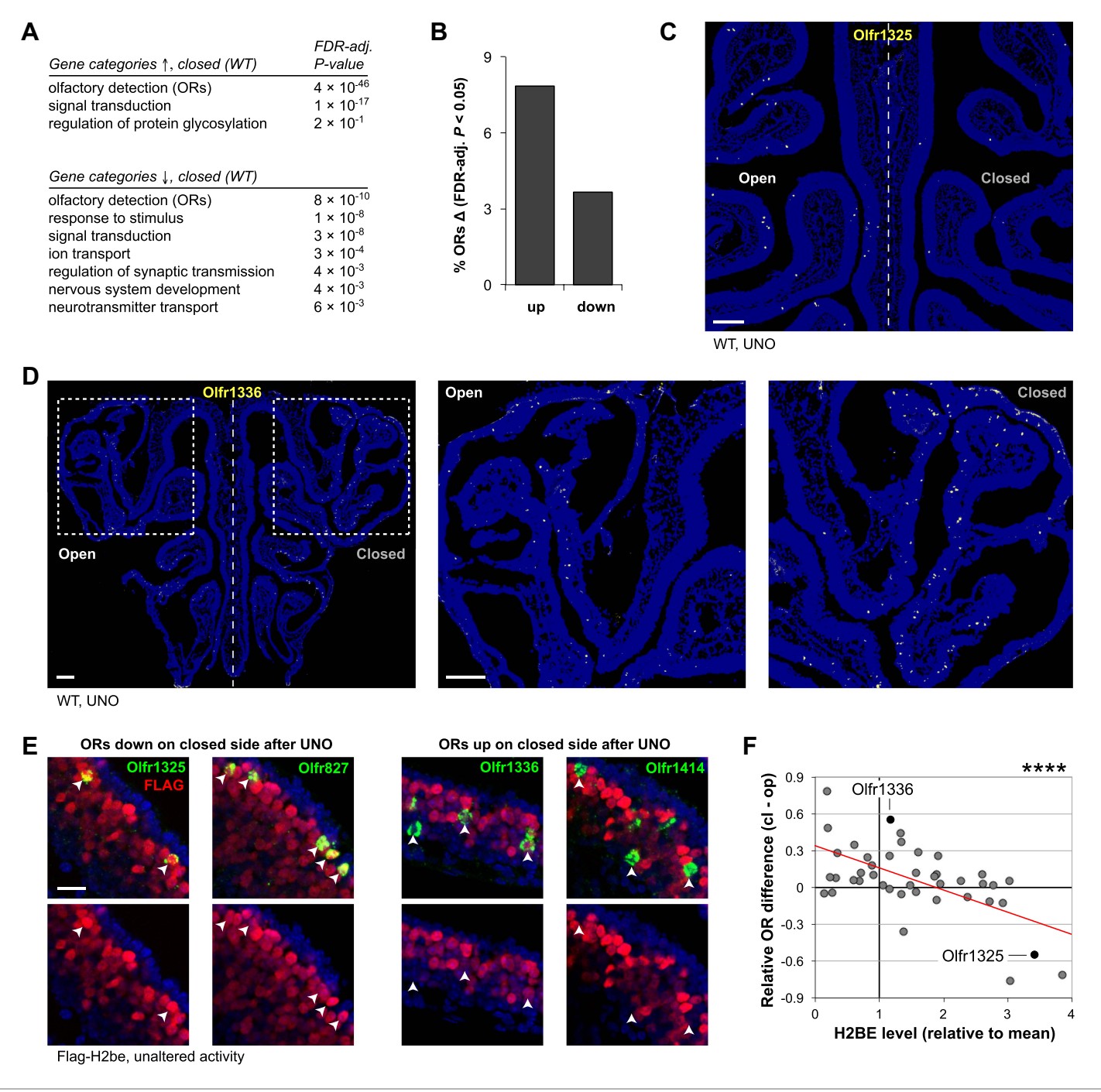

**Figure 10**. Unilateral naris occlusion (UNO) alters gene expression and OR expression frequencies. (**A**) Gene ontology (biological process) terms enriched at the top of a gene list ranked descendingly according to differential expression on the two MOE halves from 5-week old WT mice subjected to UNO (21 days), based on microarray analysis ($n = 3$ samples per MOE side, four animals per sample). (**B**) Percentage of OR genes with significantly differential (FDR-adjusted p<0.05) expression on the two sides of the MOE of WT mice after UNO (21 days; values from microarray data). (**C,D**) Representative images of *Olfr1325* (C) and *Olfr1336* (D) expression in the MOE after UNO. (**E**) Representative images of normal FLAG-H2BE levels in neurons associated with ORs that are down- (left) or up-regulated (right) in frequency after olfactory deprivation. (**F**) Relationship between UNO-mediated OR gene expression differences on the two sides of the MOE (values from microarray data for WT mice subjected to UNO for 21 days) and associated FLAG-H2BE levels measured in intact mice. Red line, best fit; ****p<0.0001. Mouse ages: (C) and (D), 5 weeks; (E), 12 weeks. Scale bars for (C) and (D), 200 μm; (E), 20 μm.

*Figure 10. Continued on next page*

*Figure 10. Continued*

The following source data are available for figure 10.

**Source data 1.** Effects of *H2be* loss of function on gene expression changes in the main olfactory epithelium (MOE) as a result of activity deprivation through unilateral naris occlusion (UNO) in 5-week old mice.

H2B-Lys5-Me immunoreactivity in H2be-KO MOEs (*Figure 14D*), while over-expression of H2BE further reduces H2B-Lys5-Me staining in H2BE-positive nuclei (*Figure 14E*). Western analysis of MOE lysates from Flag-H2be mice provided a biochemical confirmation that the Lys5-Me modification is absent from the tagged H2BE (*Figure 14F*). Together, our results suggest that, unlike canonical H2B, H2BE does not undergo detectable mono-methylation at Lys5 and that, consequently, replacement of H2B by H2BE causes a direct reduction of the Lys5-Me modification in H2BE expressing cells. Interestingly, we observe an increase in the number of H2BE expressing neurons, and in H2BE level in individual cells in the MOEs of older animals (*Figure 14G–I*). This is supported by western analysis, which revealed that the overall level of H2BE in the MOE doubles from 7 weeks to 45 weeks (approximately 10 months) of age (*Figure 14J*). In addition, we observe that the mutual exclusion between H2BE and H2B-Lys5-Me dramatically increases with age (*Figure 14G–I*), indicating the gradual replacement of canonical H2B by H2BE, which proceeds to near-completion in a fraction of neurons by 36 weeks (approximately 8.5 months) of age. Further, it suggests that the accumulation of H2BE through replacement of canonical H2B proceeds at a faster pace than it's loss through neuronal turnover.

Further analyses revealed that the presence of H2BE causes a similar reduction in the level of acetylation at Lys5, indicating that the variant also receives less of this modification compared to canonical H2B (not shown). In contrast, H2BE appears to undergo higher levels of ubiquitination at Lys120, though analyses of the modification in H2be-KO and H2be-GF mice indicate that the association is correlative but not causative (not shown).

## Discussion

We have shown here that the activity-dependent replacement of canonical H2B with H2BE, an olfactory-specific histone variant, has a direct impact on the gene expression and life span of olfactory sensory neurons. These findings uncover a novel mechanism by which the sensory experience of a neuron is recorded within its chromatin to affect its transcriptional program and longevity.

The mammalian olfactory epithelium has the unusual property of persistent neuronal self-renewal throughout adult life. Thus, the repertoire of expressed ORs in the MOE is determined by the combined probabilities associated with the choice of a specific OR by olfactory neuron precursors and the subsequent longevity of those neurons. OR gene choice has been shown to obey a largely stochastic process influenced by the genomic context of local enhancers (*Mori and Sakano, 2011*). In contrast, olfactory life span appears variable and may be influenced by environmental factors such as pathogens and odorant stimulation (*Watt et al., 2004*; *Kondo et al., 2010*). Our data uncover a chromatin-based pathway in which the absence of odor-evoked activity of specific MOE neuronal populations leads to increased H2BE expression, and in turn changes in transcription and reduced neuronal life span. In addition to a pro-apoptotic role of high levels of H2BE, we cannot exclude other roles for this olfactory-specific histone variant at low or moderate levels. Indeed our data indicate that neurons that are normally highly active and therefore express low levels of H2BE increase in abundance following olfactory deprivation, an effect that is diminished in H2be-KO mice. This result may be due to a compensatory increase of the low-H2BE expressing cells following olfactory deprivation, but could also indicate that a modest level of H2BE is optimal for neuronal longevity. In addition, our findings that changes in transcription and OR frequency following unilateral naris occlusion are significantly reduced but not eliminated in H2be-KO mice indicate that H2BE plays a key role in modulating activity dependent changes, but is likely part of a larger pathway.

The transcription factor CREB has been shown to play a major role in orchestrating transcriptional changes associated with activity-dependent neuronal plasticity and survival (*Lyons and West, 2011*; *West and Greenberg, 2011*). Interestingly, a previous study showing enhanced longevity

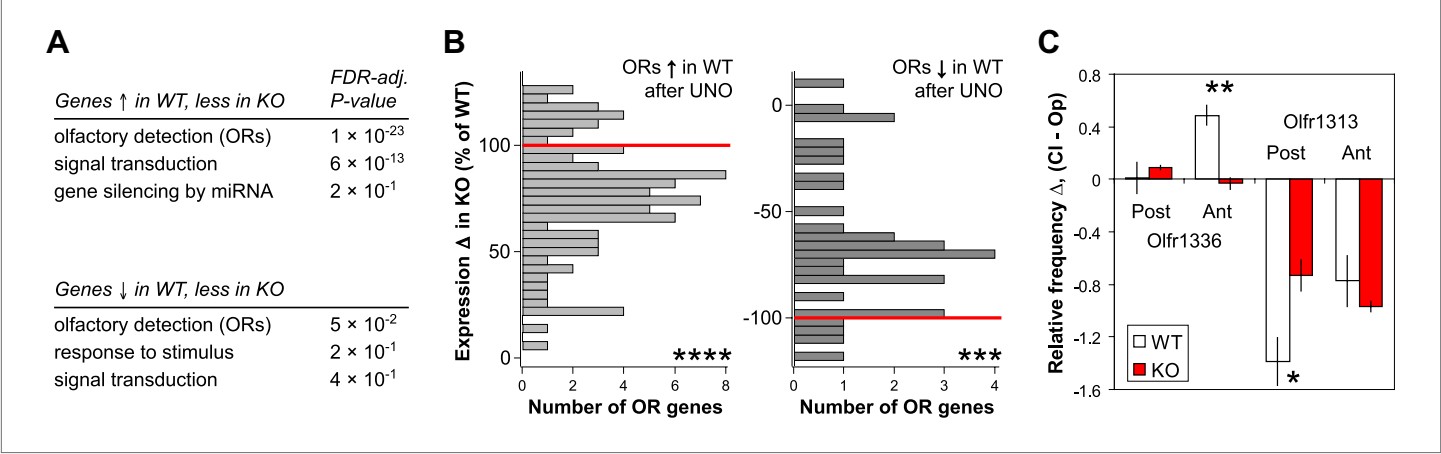

**Figure 11**. *H2be* affects activity-dependent gene expression. (**A**) Gene ontology (biological process) terms enriched among genes with UNO-mediated expression differences in WT mice ($\log_2$ fold-change > 0.3; unadjusted p<0.02), but at least 20% less altered expression in H2be-KO compared to WT mice after UNO, based on microarray analysis of MOE halves from 5-week old WT and H2be-KO mice subjected to UNO (21 days; $n$ = 3 samples per MOE side, four animals per sample). (**B**) Histograms of UNO-mediated OR expression differences on the closed and open sides of the MOE of H2be-KO mice as a percentage of the corresponding WT differences (normalized to 100% or −100%; red lines) for ORs significantly up- (left) or down-regulated (right) in WT mice after olfactory deprivation (FDR-adjusted p<0.05; values from microarray data). (**C**) Comparison of UNO-altered *Olfr1336* and *Olfr1313* frequencies in 5-week old WT and H2be-KO mice subjected to UNO (21 days). Values correspond to relative OR expression frequency differences on the two sides of the MOE according to the anterior (Ant)/ posterior (Post) position ($n$ = 3 mice, five sections per region per mouse). Note: UNO appears to affect OR frequencies differently in the anterior and posterior regions of the MOE. *p<0.05; **p<0.01***p<0.001; ****p<0.0001.

of odor-stimulated adenovirus-infected olfactory neurons implicated CREB as the mediator of this effect (***Watt et al., 2004***). The absence of full CRE sites in the *H2be* gene does not permit the establishment of a direct functional link between the activity of a CREB family member such as ICER and H2BE levels at this point, but other indirect signaling pathways may exist. An alternative scenario would postulate the existence of a cAMP-regulated chaperone that exchanges canonical H2B and H2BE.

In addition to odor-stimulated neuronal activity, olfactory neurons have been shown to display heterogeneous levels of ligand-independent, OR-derived basal activity that vary according to OR identity and are critical for regulating cAMP signals involved in axon guidance (***Imai et al., 2006***; ***Mori and Sakano, 2011***). Such studies suggest the possibility that ligand-independent, OR-derived basal signaling may also contribute to the overall activity level in mature olfactory neurons, especially under laboratory housing conditions where the odor repertoire is minimal. This scenario would help explain the observation that loss of *Adcy3*, which eliminates both basal and odor-stimulated activity, appears to affect *H2be* levels more dramatically than olfactory deprivation through UNO, which is expected to only eliminate odor-evoked activity. Thus, along with odor-evoked activity, OR-derived basal activity may be a significant contributor to the control of H2BE levels and, in turn, of olfactory neuronal longevity.

H2BE joins a list of molecules with known activity-dependent expression in olfactory neurons, many of which have roles in axon guidance and refinement. These include NRP1, KIRREL2, and EPHA5, which are up-regulated, and SEMA3A, KIRREL3, and EFNA5, which are down-regulated by neuronal activity (***Imai et al., 2006***; ***Serizawa et al., 2006***). Like SEMA3A, H2BE levels are reduced via cAMP-dependent/ $Ca^{2+}$-independent signaling, but unlike the other known activity dependent molecules, which are generally expressed in either immature or mature neurons, H2BE is expressed in both. The unusual regulation of H2BE expression likely reflects the variant's unique function in olfactory neurons.

How do the five amino acid differences between H2BE and H2B convey such distinct functional attributes? Studies of H3.3, a histone variant that differs with canonical H3 by merely four amino acids and that plays a critical role in embryonic development and gene expression in adulthood, illustrate how small sequence variations in histones can generate distinct functions (***Elsaesser et al.,***

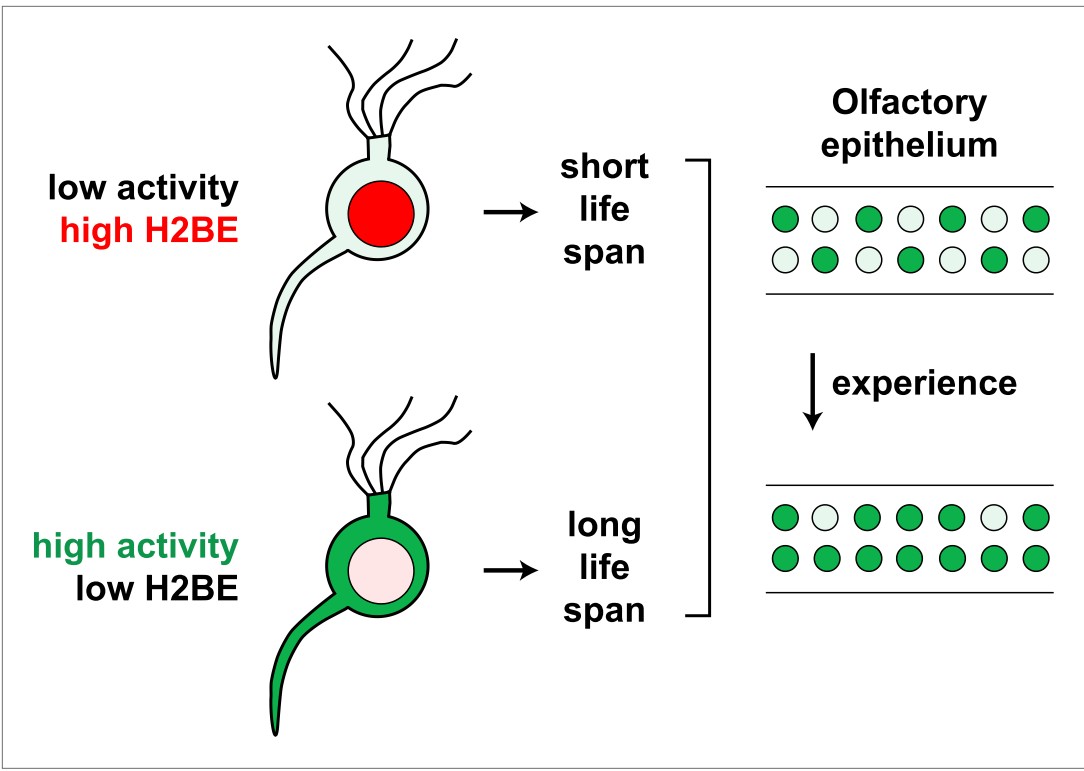

**Figure 12**. Model for the effects of neuronal activity on H2BE expression level, life span and resulting neuronal representation.

*2010*). Our analyses of relative PTM levels for H2BE and canonical H2B at a handful of known PTM sites suggest that incorporation of the variant could affect cellular transcription at least in part via differential post-translational modifiability. It must be acknowledged, however, that as is true for most histone PTMs, evidence that the PTMs affected by H2BE replacement play a role in transcription is merely correlative and therefore insufficient to support a strong functional prediction. Nevertheless, our results are consistent with the idea that H2BE shortens neuronal life span via changes in cellular transcription and metabolism, likely over the time course of several days or weeks, although the precise mechanism for this process remains to be determined. Notably, although phosphorylation of H2B Ser14 has been associated with short trigger of apoptosis in mammalian cells (*Cheung et al., 2003*), we have not found evidence for involvement of H2BE in this pathway (not shown).

The extensive activity-dependent shifts in the OR repertoire that we observe complement previous studies reporting experience-dependent sensitivity enhancements (*Hudson, 1999*) and changes in the sensory neuron representation within the MOE (*Jones et al., 2008*). A scenario thus emerges according to which neurons expressing ORs associated with environmentally salient odors are frequently active and may increase in relative abundance over time due to enhanced longevity, while neurons expressing infrequently activated ORs have a shortened life span, mediated in part by H2BE, and become less abundant (*Figure 12*). Differential longevity among olfactory sensory may provide an effective mechanism by which individuals with similar genomes adapt to diverse olfactory environments, facilitating enhanced sensitivity to odors important for survival. Accordingly, we observed significant impairment of olfactory learning behavior in H2be-KO mice, although it remains to be determined whether these defects are due to aberrant OR expression frequencies resulting from the lack of H2BE expression or, alternatively, to the altered expression of other genes involved in olfactory neuron signaling.

Among the more than 30 histone variants encoded in the mouse genome (*Marzluff et al., 2002*) only a handful have so far been characterized in terms of expression and function. Known functions of

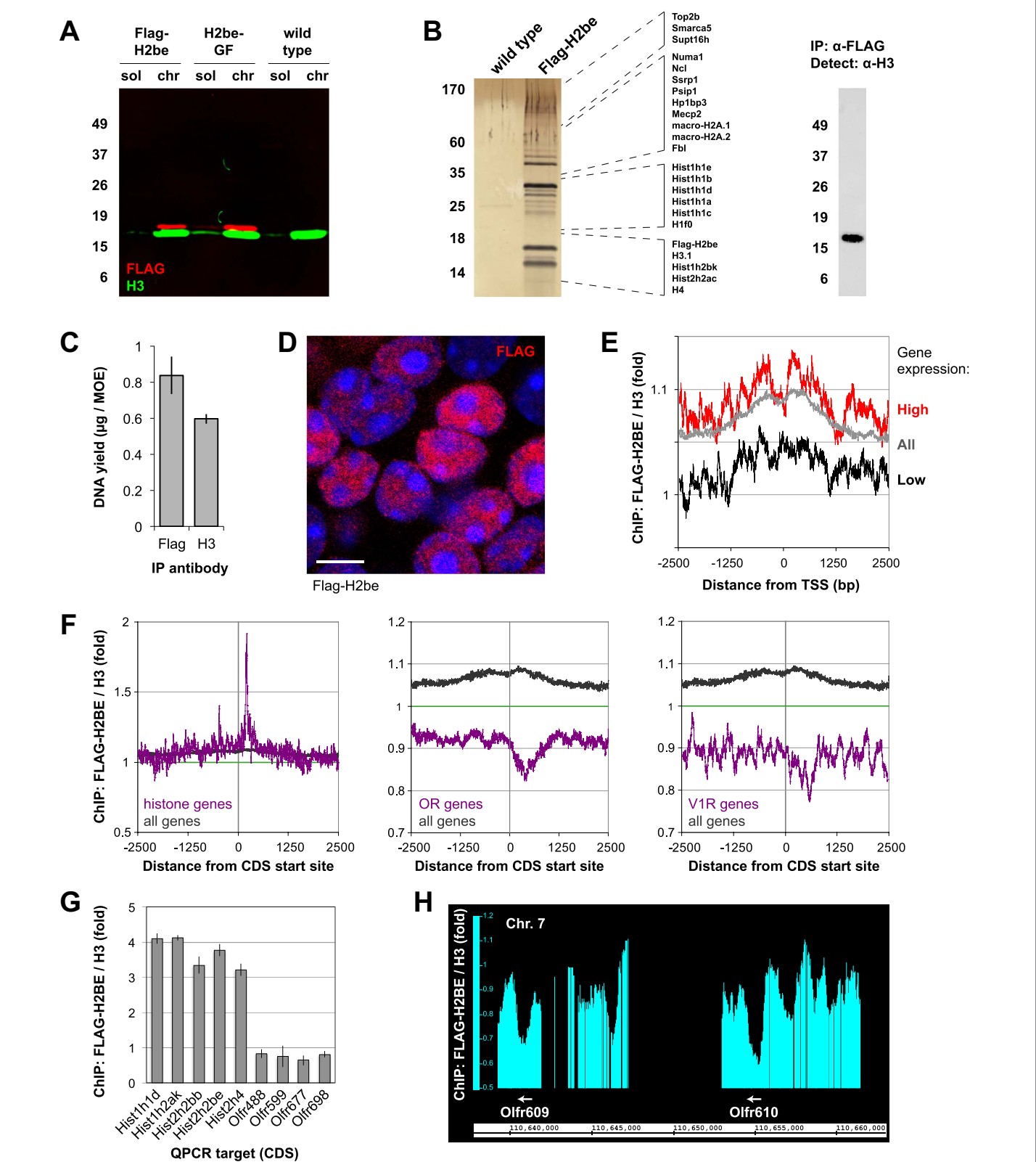

**Figure 13**. Chromatin incorporation and localization of FLAG-H2BE. (**A**) Two-color western analysis of FLAG-H2BE and H3 in soluble nucleoplasm (sol) and chromatin (chr) fractions of unfixed MOE cell nuclei from 16-week old Flag-H2be, H2be-GF, and WT mice. (**B**) SDS-PAGE analysis (left), mass spectrometric identification (listed, middle), and western analysis (right) of proteins associated with immunoprecipitated FLAG-H2BE-containing native mononucleosomes from MOE tissue of Flag-H2be transgenic mice. Proteins identified by mass spectrometry are listed according to their approximate

*Figure 13. Continued on next page*

Figure 13. Continued

electrophoretic mobility. (**C**) Quantification of DNA immunoprecipitated from crosslinked and fragmented chromatin derived from MOE tissue of Flag-H2be transgenic mice. (**D**) Representative image of FLAG-H2BE localization within olfactory neurons of a 10-week old Flag-H2be mouse. Scale bar, 5 μm. (**E**) Genome-wide ChIP analysis of relative FLAG-H2BE levels with respect to distance from the transcript start sites (TSS) for all mouse genes (grey), and genes expressed at high (red) and low (black) levels in olfactory neurons. (**F**) Genome-wide ChIP analysis of relative FLAG-H2BE levels with respect to distance from the CDS start sites for mouse histone (left), OR (middle), and vomeronasal type 1 receptor (V1R; right) genes in comparison to all genes. (**G**) Quantitative PCR analysis of relative FLAG-H2BE levels in the protein-coding regions of representative histone and OR genes. Analyses were performed on ligation-mediated-PCR-amplified Flag-H2BE and H3 ChIP DNA samples. (**H**) Genome-wide ChIP analysis shows depleted FLAG-H2BE levels within representative OR CDS regions.

histone variants include the modulation of transcription, DNA repair, meiotic recombination, chromosome segregation, sex chromosome condensation and sperm chromatin packaging (*Banaszynski et al., 2010*; *Talbert and Henikoff, 2010*). Further characterization of histone variants in the brain and in developing and self-renewing tissues represents an exciting area of future investigation.

## Materials and methods

All procedures involving animals were carried out in accordance with NIH standards and approved by the Harvard University Institutional Animal Care and Use Committee (IACUC). Unless otherwise indicated, values are presented as the mean ± standard error of the mean.

### Transgenic and gene targeted mice

The Flag-H2be transgenic mouse line (*Figure 2A*), which expresses FLAG-H2BE under control of the *H2be* promoter, was generated based on a described protocol (*Yang et al., 1997*). Briefly, a FLAG-encoding DNA sequence was inserted through homologous-recombination immediately upstream of the *H2be* CDS within BAC RP23-16G3, which contains a 200-kb region of mouse genomic sequence surrounding the *H2be* gene. The modified BAC was amplified in *E. coli*, confirmed by sequencing, and injected (Harvard Genome Modification Facility) into fertilized mouse zygotes. Transgenic founders were crossed to C57Bl/6 mice to establish the Flag-H2be line. Heterozygous Flag-H2be mice contain a single genomic copy of the transgene that is expressed in a pattern and at a level indistinguishable from that of the endogenous gene (see *Figure 2B–E*).

The H2be-KO mouse line (*Figure 3A*), in which the endogenous *H2be* CDS is replaced with a sequence encoding GAP43-mCherry (an N-terminal fusion of the first 20 amino acids of GAP43 to mCherry), was generated through homologous recombination of the endogenous *H2be* locus in mouse embryonic stem cells (ESCs) using standard methods. Following selection, ESCs were screened for the desired recombination events, confirmed by sequencing, and injected (Harvard Genome Modification Facility) into mouse blastocysts. Founders were crossed to C57Bl/6 mice to establish the H2be-KO line, in which *Gap43-mCherry* is expressed in a pattern indistinguishable from that of *H2be* (see *Figure 3B*).

The H2be-GF transgenic mouse line (*Figure 5A*), which expresses Flag-H2be under control of the olfactory marker protein (*Omp*) promoter, was generated by complete replacement of the *Omp* CDS in plasmid pJOMP (*Danciger et al., 1989*) with a sequence encoding FLAG-H2BE, followed by pronuclear injection (Harvard Genome Modification Facility) of the linearized construct into fertilized mouse zygotes. Transgenic founders were crossed to C57Bl/6 mice to establish the H2be-GF line. Heterozygous mice contain approximately 12 genomic copies of the transgene, which are expressed in mature olfactory neurons throughout the MOE with the exception of a band of neurons near zone 2 (see *Figure 5B*), a pattern that was reproducibly observed in all individuals examined (*n* = 8).

The P2-IRES-Tau-LacZ mouse line (*Mombaerts et al., 1996*), which was used to identify possible axon guidance defects in H2be-KO mice (*Figure 7B*), and the Cnga2-null/Tau-LacZ (*Zhao and Reed, 2001*) and Adcy3-null (*Wong et al., 2000*) mouse lines, which were used to identify second messengers affecting *H2be* expression (*Figure 9*) were described previously.

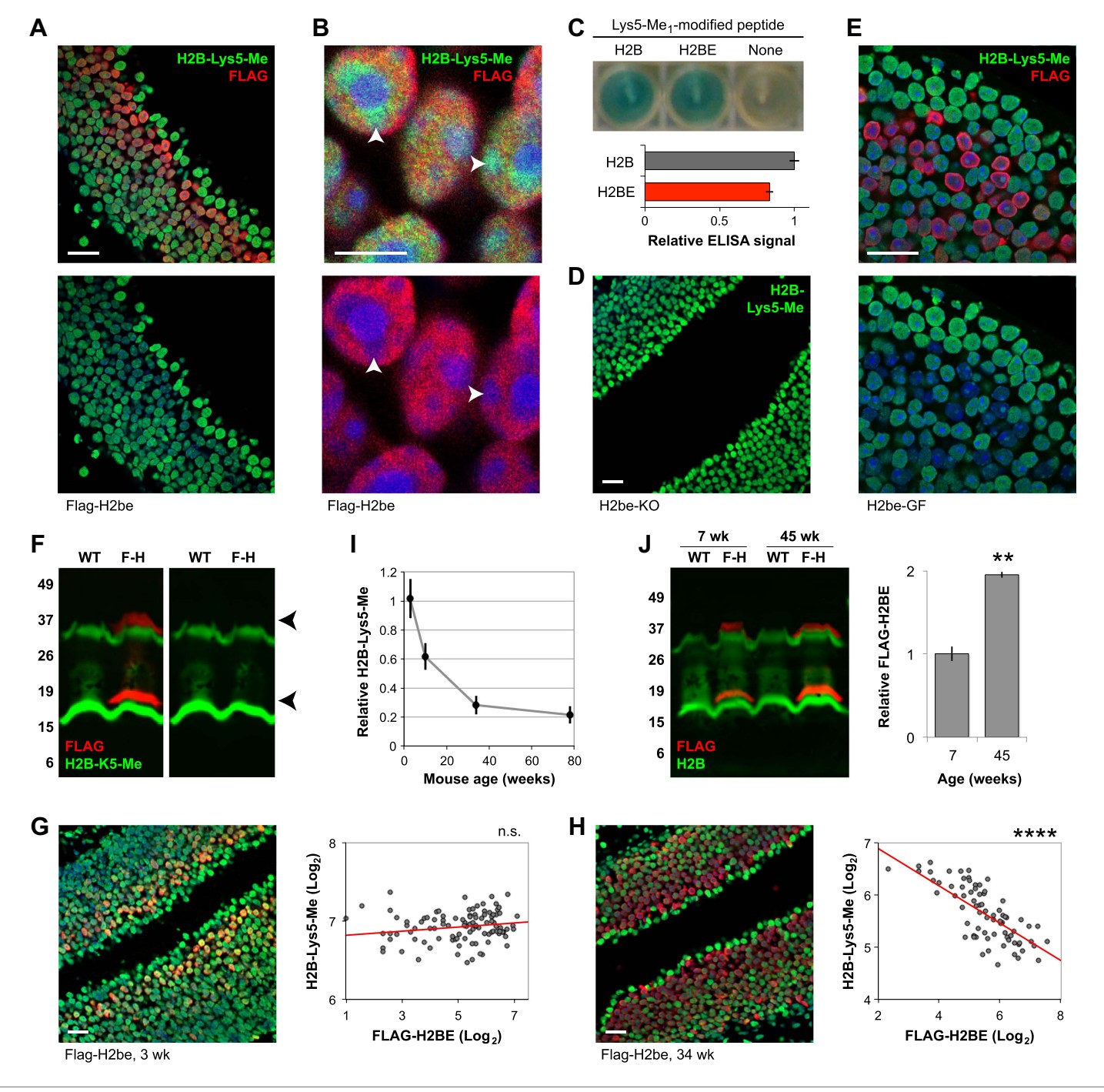

**Figure 14**. H2BE's post-translational modifications (PTMs) differ from those of canonical H2B. (**A**,**B**,**D**,**E**) Representative images of H2B-Lys5-Me (A, B, D, and E) and FLAG-H2BE (A, B, and E) staining in the MOE of Flag-H2be (A and B), H2be-KO (D) or H2be-GF (E) mice. (B) High-magnification image of FLAG-H2BE and H2B-Lys5-Me colocalization shows that H2BE is depleted in nuclear regions enriched for H2B-Lys5-Me (arrowheads). Mouse ages: (A), (B), and (E), 10 weeks; (D), 34 weeks. (**C**) Confirmation of reactivity of the anti-H2B-Lys5-Me$_1$ polyclonal antibody with the Lys5-Me PTM in the context of the H2BE protein sequence. Image (top) and quantification (bottom) of an ELISA assay for peptides corresponding to canonical H2B or H2BE and containing the Lys5-Me PTM. (**F**) Two-color fluorescent western analysis of Lys5-Me modification of FLAG-H2BE in MOE lysates from WT and Flag-H2be (F-H) mice. No detectable H2B-Lys5-Me staining of the FLAG-H2BE bands (red; arrowheads) is observed. Approximate molecular weights (kDa) are indicated (left). The bands observed at approximately 30–35 kDa likely correspond to histone dimers. (**G**–**I**) Age-dependence of H2BE accumulation. (G and H) Images (left) and quantification (right) of FLAG-H2BE and H2B-Lys5-Me co-localization in 3- (G) and 34-week old (H) mice. Red lines, best fits.
*Figure 14. Continued on next page*

*Figure 14. Continued*

(I) Quantification of H2B-Lys5-Me levels in high-H2BE neurons (relative to apical sustentacular cells; *n* = 20 nuclei from two images per timepoint). (**J**) Two-color fluorescent western analysis (left) and quantification (right) of FLAG-H2BE relative to total H2B as a function of age in MOE lysates from WT and Flag-H2be (F-H) mice. Approximate molecular weights (kDa) are indicated (left). The bands observed at approximately 30–35 kDa likely correspond to histone dimers. **$p<0.01$; ****$p<0.0001$; n.s., not significant. Scale bar for (A), (D), (E), (G), and (H), 20 µm; (B), 5 µm.

## Primary antibodies used

Active-CASP3 (rabbit polyclonal; Promega, Madison, WI, USA; G7481)
β-GAL (rabbit polyclonal; MP Biomedicals, Solon, OH, USA; 55976)
BrdU (mouse monoclonal Bu20a; Dako North America, Carpinteria, CA, USA; M0744)
FLAG (mouse monoclonal M2; Sigma-Aldrich, St. Louis, MO, USA; F1804)
FLAG (rabbit polyclonal; Sigma-Aldrich; F7425)
GAP43 (rabbit polyclonal; Novus Biologicals, Littleton, CO, USA; NB300-143A4)
H2B (rabbit polyclonal; EMD Millipore, Billerica, MA, USA; 07-371)
H2B-Lys5-Me (rabbit polyclonal; Abcam, Cambridge, MA, USA; ab12929)
H2B-Lys5-Ac (rabbit monoclonal; Abcam; ab40886)
H2B-Lys120-Ub (mouse monoclonal NRO3; Medimabs, Montréal, Québec, Canada; MM-0029)
Histone H3 (rabbit polyclonal; Abcam; ab1791)
NEUROD1 (goat polyclonal; Santa Cruz Biotechnology, Santa Cruz, CA, USA; sc1084)
OMP (goat polyclonal; Wako Chemicals USA, Richmond, VA, USA; 344-10001)
Tyrosine hydroxylase (rabbit polyclonal; Millipore; AB152)

## Histological procedures

All histological OR gene expression analyses were performed using fluorescent in situ hybridization (ISH). With the exception of *H2be* mRNA analyses (***Figure 1A,B***), which were performed using chromogenic ISH, all other histological analyses were performed using immunofluorescence (IF) or immunohistochemistry (IHC; for Active-CASP3, ***Figure 7C,E***). Unless noted, all images are of coronal tissue sections.

## Preparation of ISH probes

ISH target sequences were amplified by PCR and inserted into the pCRII-TOPO vector (Life Technologies, Grand Island, NY, USA). OR antisense probes were designed to span 500–1000 base pairs, to target CDS or UTR gene regions, and to have <70% identity to any other sequence in the mouse genome. Probes were generated from 1 µg of linearized plasmid template using T7 or Sp6 polymerases (Promega) and digoxigenin or fluorescein RNA labeling mixes (Roche Applied Science, Indianapolis, IN, USA), treated with DNaseI (Promega) and ethanol precipitation, and dissolved in a 30-µL volume of water.

## Chromogenic ISH

Whole tissues were carefully dissected from surrounding bones, frozen immediately in OCT compound (Sakura Finetek USA, Torrance, CA, USA) on dry ice, and stored at −80°C. Tissue blocks were cut into 12-µm thick cryo-sections, placed onto slides, and stored at −80°C. Chromogenic ISH experiments were performed essentially as described (***Schaeren-Wiemers and Gerfin-Moser, 1993***).

## One-color fluorescent ISH

Fluorescent ISH experiments were performed using a modified version of the chromogenic ISH method. Briefly, slide-mounted sections were warmed (37°C, 10 min), equilibrated in phosphate-buffered saline (PBS; pH 7.2; 5 min, room temperature [RT]), fixed in paraformaldehyde (PFA; 4% in PBS; 10 min, RT), washed in PBS (3 min, RT), permeabilized with Triton-X-100 (0.5% in PBS; 10 min, RT) followed by sodium dodecyl sulfate (1% in PBS; 5 min, RT), washed in PBS (3 × 3 min, RT), incubated in acetylation solution (triethanolamine [0.1 M; pH 7.5], acetic anhydride [0.25%]; 10 min, RT), washed in PBS (3 × 3 min, RT), incubated in hybridization solution (formamide [50%], SSC [2×], Denhardts [5×], yeast tRNA [250 µg/mL], herring sperm DNA [200 µg/mL], EDTA [1 mM], sodium phosphate [0.05 M; pH 7]; 30 min, RT), hybridized with a digoxigenin-labeled antisense RNA probe (1:1000 in hybridization solution; 16 hr, 42°C), washed with SSC (2×; 5 min, 42°C), washed with SSC

(0.2×; 3 × 30 min, 42°C), incubated in $H_2O_2$ (3% in TN [Tris–HCl (0.1 M; pH 7.5), 0.15 M NaCl]; 30 min, RT), washed in TNT (Tween-20 [0.05%] in TN; 3 × 3 min, RT), incubated in TNB (Blocking Reagent [Perkin Elmer, Waltham, MA, USA; 0.05% in TN]; 30 min, RT), incubated with anti-digoxigenin-POD antibody (Roche; 1:1000 in TNB; 12 hr, 4°C), and washed in TNT (3 × 20 min, RT). Fluorescent signals were generated using the Tyramide Signal Amplification (TSA) Plus Fluorescein Kit (Perkin Elmer) according to the manufacturer's instructions. Slides were mounted using Vectashield (Vector Laboratories, Burlingame, CA, USA) containing DAPI (5 µg/mL).

## Two-color fluorescent ISH

Two-color ISH was performed as described for one-color ISH, with the following modifications: Tissue sections were simultaneously hybridized with both digoxigenin- and fluorescein- or dinitrophenyl-labeled antisense RNA probes (1:1000 each in hybridization solution). Following incubation in TNB (30 min, RT), sections were incubated with anti-fluorescein-POD antibody (Roche; 1:1000 in TNB; 12 hr at 4°C) or anti-dinitrophenyl-HRP antibody (Perkin Elmer; 1:350 in TNB; 3 hr at 25°C) and washed in TNT (3 × 20 min, RT). Fluorescent signals corresponding to the fluorescein- or dinitrophenyl-labeled probes were generated using the TSA Plus Fluorescein Kit, after which sections were washed in TNT (2 × 3 min, RT), incubated in $H_2O_2$ (3% in TN; 1 hr, RT), washed in TNT (3 × 3 min, RT), incubated with anti-digoxigenin-POD antibody (1:1000 in TNB; 12 hr, 4°C), and washed in TNT (3 × 20 min, RT). Fluorescent signals corresponding to the digoxigenin-labeled probe were generated using the TSA Plus Cyanine5 Kit (Perkin Elmer) according to the manufacturer's instructions. Slides were mounted using Vectashield containing DAPI (5 µg/mL).

## Combined ISH and IF

Combined ISH and IF experiments were performed as described for one-color ISH, with the following modifications: Acetylation, which dramatically reduces detection of the FLAG epitope, was omitted. Following incubation in TNB (30 min, RT), sections were incubated with a mixture of anti-digoxigenin-POD and mouse anti-FLAG antibodies (each 1:1000 in TNB; 12 hr, 4°C) and washed in TNT (3 × 20 min; RT). Fluorescent signals corresponding to the digoxigenin-labeled RNA probe were generated using the TSA Plus Fluorescein Kit, after which sections were washed in TNT (2 × 3 min, RT), incubated with anti-mouse-Alexa647 antibody (Invitrogen; 1:1000 in TNB; 12 hr, 4°C), and washed in TNT (3 × 20 min, RT). Slides were mounted using Vectashield containing DAPI (5 µg/mL).

## One- or two-color IF

Animals were anesthetized with ketamine and perfused transcardially on ice with ice-cold PBS (25 mL) followed by ice cold PFA (4% in PBS; 25 mL). Whole tissues were carefully dissected from surrounding bones and immersed in ice-cold PFA (4% in PBS; 1 hr [OB] or overnight [MOE]). MOE tissue was decalcified in EDTA (250 mM in PBS, pH 8.5; 2 days, 4°C), and all tissues were cryoprotected in sucrose (10, 20, and 30% in PBS; 2 hr, 2 hr, and overnight, respectively). Tissues were frozen in OCT on dry ice and stored at −80°C. Tissue blocks were cut into 12-µm thick cryo-sections, placed onto slides, and stored at −80°C.

IF experiments were performed as follows: briefly, slide-mounted sections were warmed (37°C, 10 min), equilibrated in PBS (5 min, RT), fixed in PFA (4% in PBS; 10 min, RT), washed in PBS (3 min, RT), permeabilized with Triton X-100 (0.5% in PBS; 10 min, RT) followed by SDS (1% in PBS; 5 min, RT; omitted if preservation of intrinsic mCherry was necessary), washed in TNT (3 × 5 min, RT), blocked in fetal bovine serum (FBS; 10% in TN; 30 min, RT), incubated with primary antibodies (typically diluted 1:500–1:1000 in 10% FBS; 12 hr, 4°C), washed in TNT (3 × 5 min, RT), incubated with secondary antibodies (typically, Alexa488-labeled [or Alexa488- and Alexa647-labeled, for two-color IF]; Invitrogen; 1:1000 in 10% FBS; 12 hr, 4°C), and washed in TNT (3 × 15 min, RT). Slides were mounted using Vectashield containing DAPI (5 µg/mL).

## Combined IHC (active-CASP3) and IF (OMP)

Mice were perfused and the MOE tissue processed as described for IF with the following modifications: After fixation in PFA and PBS washes, slide-mounted sections were permeabilized with Triton X-100 (0.5% in PBS; 30 min, RT), washed with PBS (3 × 3 min, RT), incubated in $H_2O_2$ (3% in TN buffer; 30 min, RT), washed with TNT (3 × 3 min, RT), blocked in TNB (30 min, RT), incubated with a mixture of anti-active-CASP3 and anti-OMP antibodies (each 1:300 in TNB; 12 hr, 4°C), washed with TNT (3 × 3 min, RT), and

incubated with anti-rabbit-HRP (Jackson; 1:500 in TNB) for 12 hr, 4°C. Fluorescent signals corresponding to active-CASP3 were generated using the TSA Plus Fluorescein Kit, after which sections were washed in TNT (3 × 5 min, RT), incubated with anti-goat-Cy5 (Jackson ImmunoResearch Laboratories, West Grove, PA, USA; 1:500 in TNB; 12 hr, 4°C), and washed in TNT (3 × 15 min, RT). Slides were mounted using Vectashield containing DAPI (5 µg/mL).

## BrdU staining

Mice were injected intraperitoneally with BrdU (3 × 50 mg/kg in PBS; injections spaced 30 min apart) and sacrificed at the indicated timepoints (*Figure 7D,F*). The T = 0 timepoint, defined as 15 days post-injection, was chosen to avoid analysis of immature neurons, a large fraction of which are known to die prior to maturity (*Kondo et al., 2010*).

Mice were perfused and the MOE tissue processed as described for IF with the following modifications: After permeabilization with Triton X-100 and SDS, slide-mounted sections were washed with PBS (3 × 3 min, RT) and water (3 min at RT), incubated in HCl (2 N; 1 hr, 37°C), and washed with TNT (3 × 3 min, RT). Sections were blocked and further processed as described. Primary antibodies were used at concentrations of 1:50 (BrdU) and 1:300 (OMP).

## Imaging and quantitative fluorescence microscopy

Images were obtained using LSM710 and AxioImager Z2 (Carl Zeiss, Oberkochen, Germany) microscopes. Confocal images (1–5-µm thick optical sections) were used to quantify fluorescence signals, with care taken to ensure that exposures not exceed the instrument's dynamic range. Intensities were quantified using Zen software (Zeiss). For quantification of nuclear fluorescence in the MOE, circular regions encompassing individual nuclei were defined by DAPI fluorescence. Within each quantified image, a region surrounding the neuronal population (excluding immature and sustentacular cells) was defined to allow normalization to average neuronal nuclear fluorescence. For quantification of fluorescence in the OB, circular regions encompassing individual glomeruli were defined based on surrounding periglomerular cells, which were identified by morphology and DAPI fluorescence. p-Values corresponding to relative H2BE expression variances associated with specific ORs were calculated using a one-tailed *F*-test with FDR correction for multiple comparisons (*Benjamini and Hochberg, 1995*).

## Quantification of OR expression, apoptosis, and BrdU frequencies in the MOE

Fluorescent olfactory neuron counts corresponding to MOE tissue from an individual mouse were determined from a series of 10–12 stained coronal sections located approximately 400 µm apart and spanning the anterior–posterior length of the organ. Fluorescent cell counting was performed using Velocity software (Perkin Elmer) or, when necessary due to difficulties in resolving individual olfactory neurons, manually. Epithelial volumes were calculated from areas determined using Velocity software, based on OMP and DAPI signals.

## Analysis of the effects of *H2be* or Flag-H2be expression in cell culture

Coding sequences for H2BE, FLAG-H2BE, and consensus H2B were inserted into the pLNCX2 vector (Clontech Laboratories, Mountain View, CA, USA). Retroviruses carrying the resulting clones were generated and used according to the Retroviral Gene Transfer and Expression User Manual (Clontech). NIH-3T3 and HEK-293 cells were transduced by retroviral infection and cell lines stably expressing high levels of each transgene were selected using Geneticin (500 µg/mL; Invitrogen). Expression of transgenes in selected lines was verified by quantitative RT-PCR. Transgenic and non-transgenic cell lines were analyzed for cell viability using a Vi-Cell XR Cell Viability Analyzer (Beckman Coulter, Brea, CA, USA).

## Laser-capture micro-dissection and microarray gene expression analyses

For experiments leading to the initial identification of *H2be*, RNA was obtained by laser-capture micro-dissection (LCM) of apical and basal neurons in the VNO. Briefly, whole VNOs from 8-week old CD1 male mice were carefully dissected from surrounding bones and immediately frozen in OCT. Tissue

blocks were cut into 12-µm thick cryo-sections, placed alternately onto Superfrost and Superfrost plus slides (VWR, Radnor, PA, USA). Sections on Superfrost plus slides were stained by ISH for the Gnai2 and Gnao genes, which mark the apical and basal zones, respectively, and used as guides for LCM. Sections on Superfrost slides were stained with toluidine blue and used for LCM of approximately 100 apical and 100 basal neurons per section using a PixCell II LCM system (Arcturus, now Life Technologies, Grand Island, NY, USA). Samples were immediately frozen and pooled into groups of 10 (approximately 1000 cells per group), from which the RNA was extracted using the Arcturus Picopure Kit (Life Technologies, Grand Island, NY, USA). LCM RNA samples were amplified in parallel with whole VNO RNA samples using the Arcturus RiboAmp OA RNA Amplification Kit (Applied Biosystems) and analyzed using Mouse Genome 430 2.0 Arrays (Affymetrix, Santa Clara, CA, USA) according to the manufacturer's instructions.

For all other data reported in this study, RNA was prepared from whole MOE tissue or, in the case of UNO expression analyses, from MOE halves that had been carefully removed from the medial bone. RNA was isolated using Trizol Reagent (Invitrogen) and purified using an RNeasy Miniprep Kit (Qiagen, Valencia, CA, USA). Experiments were performed using 3–6 biological replicates per condition or genotype and MOE tissue from 2–4 individual mice per replicate. Samples were processed and applied to Affymetrix Mouse Gene 1.0 ST Arrays according to the manufacturer's instructions.

Probe cell intensity files (CEL) were analyzed for potential outliers using the Bioconductor software package arrayQualityMetrics (*Kauffmann et al., 2009*). CEL files were processed with the Affymetrix Expression Console software to generate probe level summarization files (CHP) using the iterative Probe Logarithmic Intensity Error Estimation (IterPLIER) and sketch-quantile normalization algorithms. Statistical analyses of differential expression between groups were carried out using the Bioconductor limma package (*Smyth, 2004*) implemented through the Bioconductor affylmGUI software (*Wettenhall et al., 2006*) to generate unadjusted p-values and false discovery rate (FDR) corrections for multiple comparisons. For analyses of gene expression defects in H2be-KO MOE tissue, FDR corrections were made based on all represented genes with $\log_2$ expression levels above five. For analyses of OR expression defects in H2be-GF mice and expression differences following UNO, FDR corrections were made based on all represented OR genes with $\log_2$ expression levels above seven or five, respectively.

Gene ontology analyses were performed using the GOrilla software (http://cbl-gorilla.cs.technion.ac.il/) (*Eden et al., 2007*; *Eden et al., 2009*). All analyses were carried out using the 'single ranked list of genes' mode, with the exception of the analysis summarized in *Figure 11A*, which was performed using the 'two lists of genes' mode. Reported gene ontology summaries represent a subset of biological process terms selected based on redundancy using the REViGO software (http://revigo.irb.hr/; *Supek et al., 2011*). All reported enrichment p-values are FDR-adjusted using the Benjamini–Hochberg method (*Benjamini and Hochberg, 1995*).

## Quantitative PCR (qPCR) analysis of gene expression

cDNA samples for qPCR analysis were prepared using the QuantiTect Reverse Transcription Kit (Qiagen) starting from whole MOE RNA prepared using Trizol Reagent and purified using an RNeasy Miniprep Kit. Single-plex experiments (*Figure 2D*) were performed using the QuantiTect SYBR Green PCR Kit (Qiagen) with an MJ Opticon 2 instrument (Bio-Rad Laboratories, Hercules, CA, USA). Multiplex experiments (*Figure 4B*) were performed using the QuantiTect Multiplex PCR Kit (Qiagen) with an ABI 7900HT instrument (Applied Biosystems). Primer pairs (Integrated DNA Technologies, Coralville, IA, USA) and fluorophore/quencher-labeled probes (Eurofins MWG Operon, Huntsville, AL, USA; Integrated DNA Technologies) were designed using the Primer-BLAST tool (NCBI). Primer efficiencies were assessed using standard curves and pairs exhibiting efficiencies of >99% were used for analysis.

## Unilateral naris occlusion

14-day old mice were administered Buprenorphine (0.05 mg/kg), anesthetized using isoflurane (confirmed through a tail pinch), and subjected immediately to electrocautery for approximately 5 s on the right nostril under a dissecting microscope (with care taken to avoid contact of the electrocautery unit with any non-superficial tissues). Mice were administered additional doses of Buprenorphine 12 and 24 hr after the procedure and examined on a daily basis to ensure complete blockage of the right nostril

through scar formation (typically approximately 3–5 days after the procedure) and normal mouse development and activity.

## Chronic odor exposure

Odors were presented as mixtures in mineral oil (octanal [0.31%], heptanal [0.068%], eugenol [2.65%]) or propylene glycol (lyral [10%]) continuously for 21 days at concentrations designed to achieve a vapor pressure of approximately 1 Pa per odorant. Odorants were presented in 100-μL volumes of each solution applied to a cotton pad and inserted into a metal mesh tea ball, which was suspended in the middle of the mouse cage by a metal chain. Odorants were exchanged every 24 hr.

## Nuclear fractionation, immunoprecipitation of native mononucleosomes, and mass spectrometry

Preparation of soluble nuclear proteins, chromatin, and mononucleosomes from MOE tissue and immunoprecipitation of FLAG-H2BE-containing mononucleosomes were performed as described (*Okada and Fukagawa, 2006*; *Okada et al., 2006*), with modifications. Briefly, MOE tissue from four 8-week old Flag-H2be and four 8-week old WT mice were dissected and immediately minced and mechanically homogenized. Nuclei were filtered through a cell strainer, pelleted, washed, and disrupted by sonication. Chromatin was separated from soluble nucleoplasm by centrifugation, washed, and digested with micrococcal nuclease to mononucleosomes. Mononucleosomes were solubilized in 350 mM KCl buffer and affinity-purified using Anti-FLAG M2 affinity gel (Sigma-Aldrich). Proteins were eluted with FLAG peptide (Sigma-Aldrich), TCA-precipitated, run on a 15% SDS-PAGE gel, and separated into the following molecular weight fractions: >100, 35–100, 18–35, and <18 kDa. Gel fractions were submitted for MS/MS analysis and protein identification within the Harvard University Microchemistry and Proteomics Analysis Facility.

## Genome-wide location analysis of FLAG-H2BE

Chromatin immunoprecipitation (ChIP) and ligation-mediated PCR amplification of the purified DNA were performed essentially as described (*Lee et al., 2006*). Input chromatin was prepared from whole MOE tissue dissected from 5-week old male Flag-H2be mice using mouse anti-FLAG and rabbit anti-histone H3 antibodies. Amplified DNA samples from the FLAG and H3 ChIPs were separately fragmented and analyzed in duplicate (2 mice per replicate) using GeneChip Mouse Promoter 1.0R Arrays (Affymetrix), which tile 10 kb of DNA surrounding the transcript start site of approximately 25,500 mouse genes. CEL files were processed using Tiling Analysis Software (Affymetrix) to generate BAR files.

ChIP signals were analyzed at each position in the genome as a ratio of FLAG to H3 and assigned to a specific gene promoter region using Galaxy (http://main.g2.bx.psu.edu/). Transcripts were grouped based on their expression level in OMP+ olfactory neurons, using expression values downloaded from (*Sammeta et al., 2007*), or based on their gene family. Within each group, signals at each position relative to the transcript or CDS start site were averaged and plotted to obtain signal profiles for each gene group.

Quantitative PCR analyses of relative FLAG-H2BE levels in the protein-coding (CDS) regions of histone and OR genes were performed on ligation-mediated PCR-amplified DNA from FLAG or H3 ChIP samples. Experiments were performed in triplicate for each input (FLAG or H3) and primer pair combination starting from 0.2 ng of input DNA per reaction. Reactions were performed using the QuantiTect SYBR Green PCR Kit (Qiagen) and an MJ Opticon 2 instrument (Bio-Rad). Primer pairs (Integrated DNA Technologies) were designed using the Primer-BLAST tool (NCBI). Primer efficiencies were assessed using standard curves and pairs exhibiting efficiencies of >99% were used for analysis.

## One and two-color fluorescent western blot analyses

Protein samples were separated by electrophoresis on a TRIS-glycine-SDS polyacrylamide gel (10–20% gradient; Bio-Rad), and transferred to nitrocellulose membranes. Membranes were washed in TN (5 min, RT), blocked in Blotto (Santa Cruz Biotechnology; 5% in TN; 1 hr, RT), incubated with primary antibodies (each diluted 1:2000 in 5% Blotto [in TNT]; 12 hr, 4°C), washed in TNT (3 × 5 min, RT), incubated with a mixture of Alexa488-conjugated anti-rabbit and Alexa647-conjugated anti-mouse secondary

antibodies (Invitrogen; 1:2000 in 5% Blotto [in TNT]; 1–2 hr, RT), and washed in TNT (3 × 10 min, RT). Blots were scanned using a Typhoon Trio Imager and analyzed using ImageQuant software (GE Healthcare Biosciences, Pittsburgh, PA, USA).

## ELISA

Eight-well Streptavidin High Binding Capacity strips (Thermo-Fisher Scientific, Waltham, MA, USA) were washed (Tris [25 mM, pH 7.5], NaCl [150 mM], BSA [1%], and Tween-20 [0.05%]). Custom-synthesized C-terminally-biotinylated peptides (Abgent, San Diego, CA, USA) corresponding to the N-terminal 13 amino acids of H2BE or the mouse consensus H2B and containing a methyl-modified lysine residue at position five were immobilized within the well strips (1 μg per well in block buffer [0.2 μg mouse IgG in wash buffer]; 3 hr, RT), washed, incubated with anti-H2B-Lys5-Me antibody (1:5000–1:5,000,000 in block buffer; 12 hr; 4°C), washed, incubated with anti-rabbit-HRP (1:2000 in block buffer; 1 hr, RT), and washed. ELISA signals were developed using the 1-Step Ultra TMB-ELISA reagent (Thermo Scientific) according to the manufacturer's instructions.

## Olfactory odor discrimination training

Ten 4-month old H2be-KO and heterozygous littermates were subjected to odor discrimination training using water restriction for motivation, under a behavioral paradigm similar to that described (*Uchida and Mainen, 2003*), but adapted for mice. In an initial experiment, mice were challenged to discriminate between hexanol and hexanoic acid, and in a second experiment, were challenged with the two stereoisomers of carvone. Prior to the initial experiment, mice were trained to obtain water from two ports unconditionally (day 1), after first poking an odor port (days 2–5), after poking an odor port presenting isoamyl-acetate odor (days 6–13), and after only a single poke of an odor port presenting isoamyl-acetate odor (days 14–20). Odor discrimination training began on day 21. Odors streams, controlled by an olfactometer, were generated from air passed through a filter containing 10% of the concentrated odorant in mineral oil and diluted 1:20 into a stream of clean air.

## Acknowledgements

We thank N. Uchida and M. Uchida for assistance with odor-discrimination experiments, J. Bergan for assistance with electro-olfactogram experiments, N. Rubinstein for assistance with statistical analyses, D. Storm for *Adcy3*-KO mice, and D. Reinberg, N. Francis, and members of the Dulac lab for helpful discussions.

## Additional information

### Competing interests

CD: Senior Editor, *eLife*. The other author has declared that no competing interests exist.

### Funding

| Funder | Grant reference number | Author |
| --- | --- | --- |
| National Institutes of Health | R21DC011117 | Catherine Dulac |
| National Institutes of Health | R01DC009019 | Catherine Dulac |
| Howard Hughes Medical Institute | | Catherine Dulac |
| Burroughs Wellcome Fund | 1003528.02 | Stephen W Santoro |

The funders had no role in study design, data collection and interpretation, or the decision to submit the work for publication.

### Author contributions

SWS, Conception and design, Acquisition of data, Analysis and interpretation of data, Drafting or revising the article; CD, Conception and design, Analysis and interpretation of data, Drafting or revising the article

### Ethics

Animal experimentation: This study was performed within the facilities of the Harvard University Faculty of Arts and Sciences (HU/FAS) in strict accordance with the recommendations in the Guide for the Care and Use of Laboratory Animals of the National Institutes of Health. All animals were handled according to a protocol approved by the Harvard University Institutional Animal Care and Use Committee (IACUC; protocol #97-03). The HU/FAS animal care and use program maintains full AAALAC accreditation, is assured with OLAW (A3593-01), and is currently registered with the USDA. Every effort was made to minimize animal suffering during this study.

## Additional files

### Supplementary files

• Supplementary file 1. (**A**) Effects of *H2be* loss-of-function on gene expression, based on microarray analyses of whole MOE tissue from 6-month old H2be-KO and WT mice (n = 6 samples per genotype, two animals per sample). Top 100 genes that are up- (left) or down-regulated (right) in H2be-KO mice compared to WT controls are listed according to their unadjusted p-value rank. OR genes are highlighted. (**B**) Effects of ectopic over-expression of *H2be* on gene expression, based on microarray analyses of whole MOE tissue from 5-week old H2be-GF (n = 4 samples, three animals per sample) and WT (n = 6 samples, two animals per sample) mice. Top 100 genes that are up- (left) or down- (right) regulated in H2be-GF mice compared to WT controls are listed according to their unadjusted p-value rank. OR genes are highlighted.

### Major datasets

The following datasets were generated

| Author(s) | Year | Dataset title | Dataset ID and/or URL | Database, license, and accessibility information |
| --- | --- | --- | --- | --- |
| Santoro SW, Dulac C | 2012 | Effects of *H2be* loss of function on gene expression in the main olfactory epithelium (MOE) of 6-month old mice | GSE39515; http://www.ncbi.nlm.nih.gov/geo/query/acc.cgi?acc=GSE39515 | In the public domain at GEO http://www.ncbi.nlm.nih.gov/geo/ |
| Santoro SW, Dulac C | 2012 | Effects of *H2be* loss of function on gene expression changes in the main olfactory epithelium (MOE) as a result of activity deprivation through unilateral naris occlusion (UNO) | GSE39516; http://www.ncbi.nlm.nih.gov/geo/query/acc.cgi?acc=GSE39516 | In the public domain at GEO http://www.ncbi.nlm.nih.gov/geo/ |
| Santoro SW, Dulac C | 2012 | Effects of *H2be* ectopic over-expression on gene expression in the main olfactory epithelium (MOE) of 5-week old mice | GSE39514; http://www.ncbi.nlm.nih.gov/geo/query/acc.cgi?acc=GSE39514 | In the public domain at GEO http://www.ncbi.nlm.nih.gov/geo/ |
| Santoro SW, Dulac C | 2012 | Genome-wide location analysis of FLAG-H2BE | GSE39517; http://www.ncbi.nlm.nih.gov/geo/query/acc.cgi?acc=GSE39517 | In the public domain at GEO http://www.ncbi.nlm.nih.gov/geo/ |

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
