## [Decision Letter]

Thank you for choosing to send your work entitled “The activity-dependent histone variant H2BE modulates the life span of olfactory neurons” for consideration at *eLife*. Your article has been evaluated by a Senior Editor and 2 reviewers, one of whom is a member of *eLife's* Board of Reviewing Editors.

The Reviewing Editor and the other reviewers discussed their comments before we reached this decision, and the Reviewing Editor has assembled the following comments based on the reviewers' reports. Our goal is to provide the essential revision requirements as a single set of instructions, so that you have a clear view of the revisions that are necessary for us to publish your work.

This manuscript describes thorough investigations of the expression and function of a histone H2 variant H2BE, which is specifically expressed in the olfactory and vomeronasal neurons and appears regulated by neuronal activity. H2BE is co-expressed with a subset of odorant receptors (ORs). Genetic loss-of-function and gain-of-function analyses support the model that H2BE regulates the OR expression landscape by affecting the longevity of sensory neurons. These experiments provide compelling evidence for a novel and highly interesting phenomenon and its underlying mechanisms: lack of sensory activity over a period of weeks induces the expression of H2BE, which in turn promotes the elimination of less-used OR populations. This selection allows mice to optimize the OR repertoire for their environment. The findings are also exciting for the chromatin biology field, since only few histone variants have been studied so far in a biologically relevant context.

The topic is of broad interest and the experiments are well documented. Although the key mechanism of regulating neuronal longevity by the broadly distributing H2BE in the nucleus is still unknown, this manuscript provides the basis for future studies of H2BE function. Thus, the paper should be published pending a few key experiments:

1. There is a concern whether the BAC transgenic H2BE faithfully follows the expression of the endogenous H2BE. Recently published microarray analysis of the different developmental stages of the OE shows high levels of H2BE even before the NeuroD positive stage (e.g., Nickell et al: J Comp Neurol. 2012 Aug 15;520(12):2608-29. doi: 10.1002/cne.23052). The authors should show the authenticity of the H2BE transgene expression by crossing the BAC transgenic to the mCherry knock-in allele and compare their expression pattern.

2. A question has been raised whether the Flag-H2BE is nucleosomal. Indeed histones can be incorporated in the nucleosomes in a replication independent manner but we do not know if this happens in post-mitotic olfactory neurons. In the majority of the IFs presented here H2BE forms a distinct ring on the nuclear envelope with little signal on the nucleoplasm. Thus it is possible that the transgenic H2BE is expressed too late, differentiation-wise, and cannot be incorporated in nucleosomes as efficiently as the endogenous histone. The Chip analysis does not really alleviate this concern because it uses crosslinked chromatin. A simple answer to this question would be to generate native mononucleosomes from the olfactory epithelium and show that FLAG-H2BE is incorporated in these nucleosomes, or that it can be co-immunoprecipitated with histones H3 and H4 again from native nucleosomal preparations. The same should be also done with the OMP-driven transgene. Also the authors should provide some information on the relative levels of free vs nucleosomal FLAG-H2BE in each genotype (for example they can perform salt dialysis on OE nuclei – only free histones would be extracted).

3. It is suggested that the authors take out all the information regarding the modification status of this histone variant (since it is not clear what are the effects of Lys-5 methylation in transcription anyways) or the relationship with paused or elongating polII (the images are not very convincing and they do not really provide much insight regarding the role of this histone variant). Since the biology described here is so interesting, it is better to omit preliminary or inconclusive mechanistic data and replace them with the supplemental figures from the olfactory bulb.

---

## [Author Response]

*1. There is a concern whether the BAC transgenic H2BE faithfully follows the expression of the endogenous H2BE. Recently published microarray analysis of the different developmental stages of the OE shows high levels of H2BE even before the NeuroD positive stage (e.g., Nickell et al: J Comp Neurol. 2012 Aug 15;520(12):2608-29. doi: 10.1002/cne.23052). The authors should show the authenticity of the H2BE transgene expression by crossing the BAC transgenic to the mCherry knock-in allele and compare their expression pattern*.

Based on a published report suggesting an onset of H2BE expression prior to NeuroD, the reviewers express the important concern that our H2be:Flag-H2be transgene may not fully recapitulate the endogenous H2be expression and that the endogenous gene may be initiated at an earlier developmental stage than described in our manuscript, which would potentially invalidate some of our conclusions. We have addressed this concern in multiple ways by demonstrating that (A) the transgenic and endogenous H2be genes display a similar expression onset and (B) NeuroD-positive progenitors do not yet express the endogenous H2be gene, which emerges at a later stage of olfactory neuronal differentiation. Finally, we offer a slightly different interpretation of the data in Nickell et al, which is fully compatible with our conclusions (C)

A. To directly examine the similarity of expression between the H2be:Flag-H2be transgene and the endogenous *H2be* gene, we analyzed the colocalization of FLAG-H2BE protein and *H2be* mRNA using combined IF and ISH. A representative image from this experiment (newly included as Figure 2E) demonstrates a strong agreement between cellular levels of FLAG-H2BE protein and *H2be* mRNA, despite the different cellular localization of the protein and RNA. The observation of occasional basally-located neurons in which *H2be* mRNA can be detected but FLAG-H2BE cannot is likely the expected result of a slight lag in protein production following the onset of *H2be* transcription in immature neurons. These results confirm previous expression data showing highly similar staining patterns for the FLAG-H2BE and endogenous *H2be* (e.g., Figure 1A and Figure 2B). Together with the RT-qPCR data showing identical RNA expression levels for the transgenic and endogenous alleles (Figure 2D), we believe that these data strongly suggest faithful recapitulation of endogenous *H2be* gene expression by the H2be:Flag-H2be transgene. Of note, we also performed the experiment suggested by the reviewers (crossing *the BAC transgenic to the mCherry knock-in allele and compare their expression pattern),* but have not included the data as the presence of membrane-bound mCherry in axons crossing the basal layer of the olfactory epithelium, in addition to the cell bodies, makes it impossible to obtain a clear cellular resolution of H2BE expression onset by immunostaining or intrinsic fluorescence.

B. To further confirm our previous findings of the lack of co-expression of *H2be* and *Neurod1* at the protein level (based on FLAG-H2BE protein expression in Flag-H2be transgenic mice; Figure 6B), we also examined a potential overlap between endogenous *H2be* and *Neurod1* at the mRNA level in wild type mice. A representative image of this experiment (newly included as Figure 6C) clearly demonstrates the complete absence of overlap between the expression patterns of the two mRNAs, strongly supporting our conclusion that the onset of *H2be* expression strictly follows and does not include the developmental window of *Neurod1* expression.

C. We are grateful to the reviewers for drawing our attention to the recently published Nickell et al. paper, which we think is an excellent resource for understanding comprehensive gene expression changes that occur during olfactory development. After carefully reading the Nickell paper, however, we respectfully disagree with the reviewers’ interpretation that *H2be* is highly expressed prior to the onset of *Neurod1.* Although the authors of the Nickell paper used a transgenic GFP marker (called TgN1-2G) driven from the *Ngn1* promoter, which is active in neural progenitors, they found that the GFP levels are actually highest in immature neurons, not in earlier neural progenitor cells (i.e., when *Neurod1* is expressed). As stated by Nickell et al. (page 2612):

“Even if the TgN1-2G transgene is faithful to the expression pattern of the endogenous gene, rapid transition from immediate neuronal precursor basal cells into immature OSNs could result in fluorescent labeling primarily of immature OSNs. Consistent with this expectation, we found that the GFP fluorescence pattern consisted of only a few basal cells but many cells with fluorescent dendrites and axons—cells located at the depth of the immature OSN layer of the olfactory epithelium (Fig. 1C). This GFP fluorescence overlapped well with immunoreactivity for the immature OSN-specific marker Gap43, but not for the mature OSN-specific marker, Omp, confirming that the highly fluorescent cells were nearly all immature OSNs (Fig. 1C).”

Thus, the Nickell paper shows only that *H2be* is expressed at high levels in immature neurons (i.e., GAP43+), which is consistent with our data, but it gives no insight into its expression in neural progenitors.

*2. A question has been raised whether the Flag-H2BE is nucleosomal. Indeed histones can be incorporated in the nucleosomes in a replication independent manner but we do not know if this happens in post-mitotic olfactory neurons. In the majority of the IFs presented here H2BE forms a distinct ring on the nuclear envelope with little signal on the nucleoplasm. Thus it is possible that the transgenic H2BE is expressed too late, differentiation-wise, and cannot be incorporated in nucleosomes as efficiently as the endogenous histone. The Chip analysis does not really alleviate this concern because it uses crosslinked chromatin. A simple answer to this question would be to generate native mononucleosomes from the olfactory epithelium and show that FLAG-H2BE is incorporated in these nucleosomes, or that it can be co-immunoprecipitated with histones H3 and H4 again from native nucleosomal preparations. The same should be also done with the OMP-driven transgene. Also the authors should provide some information on the relative levels of free vs nucleosomal FLAG-H2BE in each genotype (for example they can perform salt dialysis on OE nuclei – only free histones would be extracted)*.

The reviewers express an important concern about the extent to which the transgenic H2BE is nucleosome-bound versus free in the nucleoplasm. We have addressed this concern in multiple ways, and have confirmed the largely chromatin-bound state of transgenic H2BE:

A. To directly investigate the extent to which FLAG-H2BE is chromatin-bound, we isolated unfixed MOE cell nuclei from Flag-H2be and H2be-GF transgenic mice and then fractionated the nuclei into soluble nucleoplasm and chromatin, based on a protocol published by the Fukagawa lab (Okada and Fukagawa, *Protocol Exchange,* 2006). We used two-color western analysis to compare the amount of FLAG-H2BE and H3 in the two fractions. An image of the immunoblot (newly included as Figure 13A) shows that FLAG-H2BE and H3 are present at very low levels in soluble nucleoplasm and are almost entirely chromatin bound.

B. To further investigate the association of H2BE with chromatin, we immunoprecipitated native mononucleosomes containing FLAG-H2BE from chromatin of transgenic Flag-H2be mice and analyzed the associated proteins by mass spectrometry and western. This experiment (summarized in the newly included Figure 13B) revealed a large number of chromatin-associated proteins that were immunoprecipitated with H2BE at the nucleosome level, including many other histone proteins. These results provide strong evidence that FLAG-H2BE is integrated into the chromatin of olfactory neurons.

C. In performing our genome-wide location analysis experiments, we measured the amount of DNA immunoprecipitated from chromatin of transgenic Flag-H2be mice using anti-FLAG and anti-H3 antibodies. These measurements (summarized in the newly included Figure 13C) show that both antibodies precipitate large quantities of DNA. These data further confirm that FLAG-H2BE is integrated into the chromatin of olfactory neurons.

D. There are two likely explanations for the appearance of a nuclear ring in some of our IF images: 1) olfactory neurons have a single or small number of large central chromocenters that are not stained with anti-FLAG antibodies, leading to the appearance of stronger staining on the nuclear periphery (for example, see Figure 13D). 2) The anti-FLAG antibody stains chromatin within the nuclear periphery more quickly than the euchromatin within the nuclear interior. This is particularly evident for MOE tissue from H2be-GF mice, where the level of Flag-H2be is much higher than in Flag-H2be mice (for example, Figure 14E). We have observed that more uniform staining throughout the nucleus is achieved with long antibody incubation times, indicating that the apparent peripheral staining is not due to concentration of FLAG-H2BE at the nuclear periphery.

*3. It is suggested that the authors take out all the information regarding the modification status of this histone variant (since it is not clear what are the effects of Lys-5 methylation in transcription anyways) or the relationship with paused or elongating polII (the images are not very convincing and they do not really provide much insight regarding the role of this histone variant). Since the biology described here is so interesting, it is better to omit preliminary or inconclusive mechanistic data and replace them with the supplemental figures from the olfactory bulb*.

The reviewers suggest that we remove the data related to the differential PTM status of canonical and variant H2B and the data concerning the effects of H2BE on the elongation status of polII as they find them mechanistically uninformative (PTM) or weak (polII). As detailed below we agree with some of this assessment, and propose a compromise in which we keep what we view as the strongest and most important data, while downplaying some of their mechanistic implications in the text of the manuscript.

A. With respect to the findings that H2BE receives dramatically reduced monomethylation at Lys5 compared to canonical H2B, we agree with the reviewers that the biological implication is presently unknown. Although this modification has been strongly correlated with active transcription in other cell systems (Barski et al. 2007, Cell *129,* 823-837; Wang et al. 2008, Nature Genetics *40,* 897-903), a causative association has not been established. Moreover, since we did not survey all possible H2B modifications, our understanding of H2BE’s modification status is by no means complete. However, we are reluctant to entirely omit these data, as we believe that the finding represents an important proof of concept for potential functional differences between the two histones at the molecular level, which must be mediated by only five amino acid differences. We believe that the finding of reduced Lys5-Me modification serves as a platform for further investigations into the functional differences between the two molecules and possibly for understanding the biological function of the H2B-Lys5-Me modification. Thus, we wish to keep these data in the manuscript, but we have modified the text to emphasize that the finding is not an indication of mechanism.

B. With respect to our findings that H2BE causes reduced RNAPII elongation and increased pausing, we believe that the data presented are solid and significant, but we agree with the reviewer that they are less convincing, and that they be corroborated using other methods. Thus, we have removed this set of results entirely. Please note: we have eliminated or moved supplemental data to main figures, and in particular the images of the olfactory bulb to which the reviewers referred.